# Evaluating and Explaining the Severity of Distribution Shifts: Illustration with Tabular Text Classification

## Abstract

After deploying a machine learning model, distribution shifts may emerge in real-world data. When dealing with unlabeled data, it can be challenging to accurately assess the impact of these drifts on the model's performance, for any type and intensity of shift. In that case, decisions such as updating the model for every benign shift would not be cost-efficient. In this paper, we introduce the *Error Classifier*, an error assessment method that addresses two tasks: unsupervised performance estimation and error detection on out-of-distribution data. The *Error Classifier* computes the probability that the model will fail based on detected fault patterns. Further, we employ a sampling-based approximation of Shapley values, with the *Error Classifier* as value function, in order to explain why a shift is predicted as severe, in terms of feature values. As explanation methods can sometimes disagree, we suggest evaluating the consistency of explanations produced by our technique and different ones. We focus on classification and illustrate the relevance of our method in a bimodal context, on tabular datasets with text fields. We measure our method against a selection of 15 baselines from various domains, on 7 datasets with a variety of shifts, and 2 multimodal fusion strategies for the classification models. Lastly, we show the usefulness of our explanation algorithm on instances affected by various types of shifts.

## 1 Introduction

While pretrained language models such as BERT can achieve state-of-the-art performance in various tasks such as classification (Devlin et al., 2019), a mismatch between the source (training/fine-tuning) and target (test) distributions can deteriorate the model's performance (Yuan et al., 2023). Once dataset shifts have been detected based on unlabeled data (Rabanser et al., 2019), it is essential to assess the severity of their impact on the model's performance in order to make informed decisions. It would be cost-inefficient to update a model or integrate human control of the model's outputs for every benign shift. However, no method may accurately predict out-of-distribution (OOD) performance for every type and intensity of distribution shifts (Garg et al., 2022). Despite this, it would be still useful to understand which estimator is more reliable for unsupervised performance estimation and error detection across a diversity of shifts. Unsupervised performance prediction aims to evaluate a model based on unlabeled datasets whereas error detection attempts to identify mispredicted target inputs (Chen et al., 2021a). In that case, it can be valuable to explain why a shift is evaluated as severe, in terms of feature values. For instance, in medical diagnosis prediction, subject matter experts might find it useful to understand why a shift could affect the reliability of a given model's prediction, and then decide to override the initial outcome.

Here the focus is on multimodal classification tasks based on tabular datasets with text fields in English. These datasets consist of categorical and numerical features (i.e. the tabular modality) and fields with free-form text (i.e. the text modality) (Shi et al., 2021). Various critical applications rely on such datasets. In the medical field, patient characteristics and clinical notes could be employed for diagnosis prediction. In financial investment, models could make decisions based on time series (e.g. asset price) and text news for sentiment analysis.

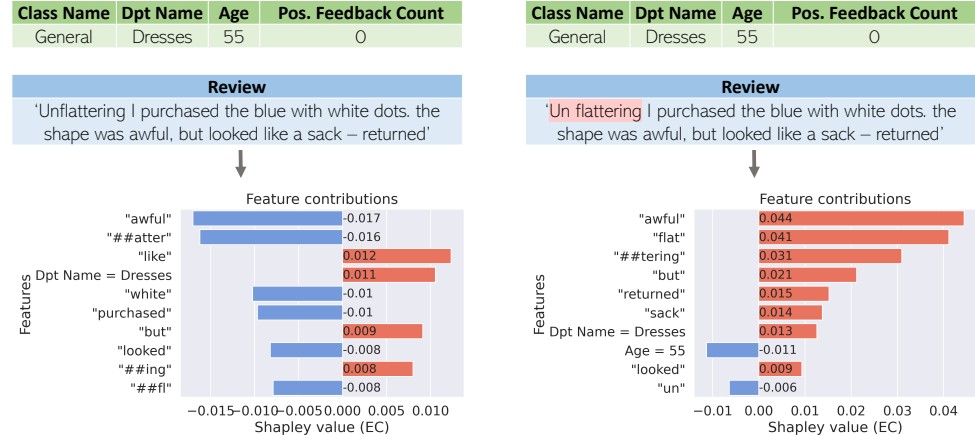

Figure 1: **Illustration of our method.** *Left*: Original multimodal input (top) with true rating of 2 for a user review regarding clothing items (sentiment analysis with cloth dataset). The classification model predicts the correct label. EC outputs a probability of error of 44%. *Right*: Shifted input with an extra space on the first word of the text field. The classification model overestimates the rating. EC estimates a probability of error of 63%. The outputs of the explanation algorithm (top 10 feature contributions with EC as value function) are displayed in each bar plot (bottom).

The tasks of unsupervised performance estimation and error detection are complex to perform across diverse distribution shifts. Our objective is to evaluate what estimator may perform best on a variety of distribution shifts that could occur in practice. We highlight the following contributions:

- We introduce the *Error Classifier* (EC), an error assessment method that can be employed for OOD performance estimation and error detection. For any new input, EC computes the probability that the classification model will fail based on fault patterns learned on calibration data.

- We show that EC can be used as value function in a sampling-based algorithm that approximates Shapley values. Once a shift has been detected and assessed, this turns out to be useful to explain why it is predicted as potentially harmful (i.e. what feature values contribute to the probability of error estimated by EC). We evaluate the quality of an explanation by verifying its consistency across various explanation methods.

- We assess our method by comparing it to 15 baselines from different domains on 7 tabular-text classification datasets, and 2 multimodal fusion schemes for the classification models. All the methods are external approaches that can be applied to pretrained networks without modification.

Figure 1 shows an example from a sentiment analysis task, where the original multimodal input (top left) has been synthetically modified with a typographical error on the first word of the text field (top right). The classification model predicts the correct rating for the first input, but overestimates it for the modified input. Our method, EC, estimates that failure is more likely for the second prediction (63%) than for the first one (44%). Further, our sampling-based algorithm displays the top 10 feature contributions to the probability of failure (bottom). In particular, the bottom right bar plot displays several positive contributions which highlight the uncertainty caused by the combination of certain tokens: e.g. "awful" and "flattering" ("flat" + "##tering") appearing in the same text. This explanation method could be useful in critical applications (e.g. financial or medical field) where subject matter experts need to understand if and why a prediction is likely to be incorrect.

## 2 PRIOR WORK

**Multimodal fusion.** A multimodal model leverages heterogeneous and connected modalities like audio, image and text as inputs. This approach aims to learn representations of cross-modal interactions by fusing information across diverse modalities (Liang et al., 2022; Xu et al., 2023). With early fusion, cross-modal interactions happen at an early stage. For a Transformer with early con-

catenation of two modalities, this means that full pairwise attention will be computed at all layers. In contrast, late fusion of final representations makes cross-modal interactions occur at a later stage.

**Distribution shifts.** Dataset shifts appear when the respective source and target joint distributions $p$ and $q$ are different: $p(\mathbf{x}, y) \neq q(\mathbf{x}, y)$ for covariates $\mathbf{x}$ and class variable $y$ (Moreno-Torres et al., 2012). Changes in the distribution of the input variables $\mathbf{x}$ are referred to as covariate shifts. Focusing on text data, Arora et al. (2021) identify two types of OOD texts: the background shift (e.g. when style of text changes) and semantic shift (e.g. when unseen classes appear at test time). Semantic features are discriminative for the prediction task while background attributes are not. With regard to multimodal distribution shifts, Qiu et al. (2022) benchmark the robustness of image-text models under multimodal perturbations.

**OOD detection and performance estimation.** Two-sample tests metrics can be used to detect dataset shifts: univariate Jensen-Shannon Distance (Lin, 1991) and multivariate maximum mean discrepancy (Gretton et al., 2012). The domain classifier (Rabanser et al., 2019) is trained to discriminate between data from source (class 0) and target (class 1) domains. A shift is detected when this model can easily identify from which domain the samples originate. Distance-based methods, such as non-parametric deep nearest neighbors (Sun et al., 2022), can leverage feature embeddings from a model in order to perform OOD detection. With respect to OOD performance estimation, the Average Thresholded Confidence technique learns a cut-off value on the source validation data so that the proportion of samples with score (e.g. maximum confidence) above that threshold matches the accuracy (Garg et al., 2022). The predicted accuracy is estimated as the fraction of target examples for which the score exceeds that threshold. With the Difference of Confidences approach (Guillory et al., 2021), the accuracy change between the source and unlabeled target data is assessed by the difference between the average confidences on these two datasets. With Mandoline, Chen et al. (2021b) estimate the target error rate by performing importance re-weighting of the 0-1 loss with slicing functions designed to capture possible axes of distribution shifts. Lastly, Yu et al. (2022) propose the Projection Norm to predict the OOD test error. First, a new neural network is trained on the test samples which have been pseudo-labeled by the in-distribution model. Then, the more the new model's parameters are different from the original model, the greater the predicted OOD error.

**Confidence scores and uncertainty.** The maximum softmax probability turns out to be a useful baseline to estimate confidence under distribution shifts (Hendrycks & Gimpel, 2017). However, as models such as neural networks can be miscalibrated, techniques such as temperature scaling are suggested to better calibrate the class probability estimates (Guo et al., 2017a). Liu et al. (2020) show the relevance of the energy score in OOD detection tasks as it is aligned with the probability density of the input. To quantify predictive uncertainty, methods such as conformal prediction can produce prediction sets based on an expected coverage level (Vovk et al., 2005; Papadopoulos et al., 2002). In particular, Tibshirani et al. (2019) propose a weighted version of conformal prediction under covariate shift. To estimate predictive uncertainty, Lakshminarayanan et al. (2017) employ deep ensembles with random parameter initialization for each neural network, along with random shuffling of the data points. The predictive entropy can be computed after averaging the predicted probabilities from each network. To avoid the computational cost of Bayesian models, Gal & Ghahramani (2016) introduce a Bayesian approximation for deep neural networks. When evaluating the predictive uncertainty for a test input, the Monte Carlo dropout corresponds to performing various forward passes with dropout. To evaluate the trustworthiness of predictive uncertainty, Ovadia et al. (2019) present a benchmark of different methods under dataset shift (e.g. deep ensembles). To explain uncertainty estimates, Antoran et al. (2021) propose CLUE, a method based on counterfactuals, which identifies which features are responsible for uncertainty in probabilistic models. This method also makes it possible to distinguish aleatoric uncertainty from epistemic uncertainty. The second type originates from the model's parameters being under-specified by the data and increases with OOD inputs. Lastly, Watson et al. (2023) explain predictive uncertainty by adapting the computation of Shapley values (Shapley, 1953) with the conditional entropy as value function.

**Error detection.** To detect model failure during inference, Corbière et al. (2019) propose a method which estimates the true class probability in image classification tasks. Self-training ensembles can be leveraged for error detection and unsupervised accuracy estimation (Chen et al., 2021a). Concerning explanation methods, Parcalabescu & Frank (2023) introduce MM-SHAP, a multimodality score based on Shapley values, which helps detect unimodal collapse. However, Krishna et al. (2022) point out that the outputs of different explanation techniques can disagree with each other, and suggest various metrics to measure disagreement between top-k features: intersection or rank.

## 3 METHODS

Assume we have a $C$-class classification problem, where each input $\mathbf{x} = (\mathbf{x}^{text}, \mathbf{x}^{tab}) \in \mathbb{X}$ contains text fields and tabular features. The true class is $y \in \mathbb{Y} = \{0, 1, \ldots, C-1\}$. We consider a Source dataset $\mathcal{S} = \{(\mathbf{x}_i, y_i)\}_{i=1}^n$, which includes $n$ points sampled i.i.d. from distribution $p$. Further, $\mathcal{S}$ is randomly partitioned into a training dataset $\mathcal{S}_{train}$ and a validation dataset $\mathcal{S}_{val}$. We consider a class of hypotheses $\mathcal{H}$ mapping $\mathbb{X}$ to $\Delta^{C-1}$, where $\Delta^{C-1}$ is the probability simplex over $C$ classes. Given a classifier $\hat{\pi} \in \mathcal{H}$ fitted on $\mathcal{S}_{train}$, the predicted label is $\hat{y} = \arg\max_{j \in \mathbb{Y}} \hat{\pi}_j(\mathbf{x}), \forall \mathbf{x} \in \mathbb{X}$. Further, the 0-1 loss is defined as $\mathcal{L}(\hat{\pi}(\mathbf{x}), y) = \mathbf{1}_{y \neq \hat{y}}$, where $\mathbf{1}_{condition}$ is 1 if the condition is true, 0 otherwise. To evaluate the performance of $\hat{\pi}$, we define the error rate on dataset $\mathcal{D}$, indexed by set $\mathcal{I}_\mathcal{D}$, as $\varepsilon_\mathcal{D} = \frac{1}{|\mathcal{I}_\mathcal{D}|} \sum_{k \in \mathcal{I}_\mathcal{D}} \mathbf{1}_{y_k \neq \hat{y}_k}$, where $|\mathcal{I}|$ denotes the cardinality of a set $\mathcal{I}$. Given $\hat{\pi}$ and unlabeled target dataset $\mathcal{T}$, our objective is to predict the error rate and identify mispredicted inputs.

### 3.1 THE ERROR CLASSIFIER

#### 3.1.1 DESIGN AND USE

The Error Classifier (EC) estimates the likelihood that $\hat{\pi}$ will fail based on detected error patterns. We first extract the feature embedding $\mathbf{z}$ from the model $\hat{\pi}$: we have $\mathbf{z} = \phi(\mathbf{x})$, where the multimodal feature encoder $\phi : \mathbb{X} \to \mathbb{R}^d$ includes a fusion scheme (e.g. late fusion), and $d$ is the embedding dimension. Then, we construct the label by computing the 0-1 loss for each data point of $\mathcal{S}_{val}$, indexed by $\mathcal{I}_{val}$. Lastly, the EC model $\hat{f} : \mathbb{R}^d \to \Delta^1$ learns to detect error patterns: $\hat{f} = \mathcal{C}(\{(\mathbf{z}_i, \mathcal{L}(\hat{\pi}(\mathbf{x}_i), y_i)) : i \in \mathcal{I}_{val}\})$, where $\mathcal{C}$ denotes any classification algorithm that takes in data indexed by $\mathcal{I}_{val}$ in order to output a classifier fitted on that data, and where $\mathbf{z}_i = \phi(\mathbf{x}_i)$.

For a new input $\mathbf{x}'$, we address the error detection task by computing $\hat{f}_1(\mathbf{z}')$, where $\mathbf{z}' = \phi(\mathbf{x}')$ and $\hat{f}_1(\mathbf{z}')$ estimates the probability that the loss equals 1 given $\mathbf{z}'$. Lastly, we address the performance estimation task on dataset $\mathcal{T}$, indexed by $\mathcal{I}_\mathcal{T}$, by computing the error rate $\hat{\varepsilon}_\mathcal{T} = \frac{1}{|\mathcal{I}_\mathcal{T}|} \sum_{k \in \mathcal{I}_\mathcal{T}} \hat{f}_1(\mathbf{z}_k)$.

#### 3.1.2 IMPORTANCE WEIGHTING PERSPECTIVE

With importance weighting (Horvitz & Thompson, 1952), it is possible to assess a function $h(\mathbf{x}, y)$ under the target distribution $q$, given $n$ samples $\{(\mathbf{x}_i, y_i)\}_{i=1}^n$ drawn from the source distribution $p$: $\mathbb{E}_q[h(\mathbf{x}, y)] = \mathbb{E}_p[\frac{q(\mathbf{x}, y)}{p(\mathbf{x}, y)} h(\mathbf{x}, y)]$. We suppose that $\hat{f}$ is a decision tree classifier (Breiman et al., 1984). The leaf nodes in a tree form a partition of the feature space; let $\lambda(\mathbf{z}_k)$ denote the set of indices of the points from $\{(\mathbf{z}_i, \mathcal{L}(\hat{\pi}(\mathbf{x}_i), y_i)) : i \in \mathcal{I}_{val}\}$ that belong to the same leaf node based on the decision rules (i.e. decision path) fulfilled by $\mathbf{z}_k$. For the performance estimation task, we have:

$$\hat{\varepsilon}_\mathcal{T} = \frac{1}{|\mathcal{I}_\mathcal{T}|} \sum_{k \in \mathcal{I}_\mathcal{T}} \hat{f}_1(\mathbf{z}_k) = \frac{1}{|\mathcal{I}_\mathcal{T}|} \sum_{k \in \mathcal{I}_\mathcal{T}} \sum_{i \in \lambda(\mathbf{z}_k)} \frac{\mathcal{L}(\hat{\pi}(\mathbf{x}_i), y_i)}{|\lambda(\mathbf{z}_k)|} \tag{1}$$

$$= \frac{1}{|\mathcal{I}_\mathcal{T}|} \sum_{k \in \mathcal{I}_\mathcal{T}} \sum_{i \in \mathcal{I}_{val}} \frac{\mathcal{L}(\hat{\pi}(\mathbf{x}_i), y_i)}{|\lambda(\mathbf{z}_k)|} \mathbf{1}_{i \in \lambda(\mathbf{z}_k)} \tag{2}$$

$$= \frac{1}{|\mathcal{I}_{val}|} \sum_{i \in \mathcal{I}_{val}} \left( \sum_{k \in \mathcal{I}_\mathcal{T}} \frac{\mathbf{1}_{i \in \lambda(\mathbf{z}_k)}/|\mathcal{I}_\mathcal{T}|}{|\lambda(\mathbf{z}_k)|/|\mathcal{I}_{val}|} \right) \mathcal{L}(\hat{\pi}(\mathbf{x}_i), y_i) \tag{3}$$

(3) can be interpreted as importance weighting of the 0-1 loss where the numerator in the term in parenthesis can be seen as a ratio of target and source probability densities. For instance, for a given $i \in \mathcal{I}_{val}$, if $\mathbf{z}_i$ and a large proportion of target samples $\mathbf{z}_k$ follow the same decision path, then summing all the $\mathbf{1}_{i \in \lambda(\mathbf{z}_k)}/|\mathcal{I}_\mathcal{T}|$ over $\mathcal{I}_\mathcal{T}$ will result in a high probability in the numerator, with common denominator $|\lambda(\mathbf{z}_k)|/|\mathcal{I}_{val}|$. Further, if the related leaf node contains few validation samples, then the corresponding probability (i.e. $|\lambda(\mathbf{z}_k)|/|\mathcal{I}_{val}|$) will be low. In that case, the corresponding loss $\mathcal{L}(\hat{\pi}(\mathbf{x}_i), y_i)$ will be assigned an important weight.

### 3.1.3 RECALIBRATION PERSPECTIVE

The probability estimates produced by classifiers should be confidence-calibrated in order to reflect the trustworthiness of the predictions (Guo et al., 2017b). Considering our classifier $\hat{\pi}$, perfect calibration is defined as $\mathbb{P}[y = \arg\max_{j \in \mathbb{Y}} \hat{\pi}_j(\mathbf{x}) \mid \max_{j \in \mathbb{Y}} \hat{\pi}_j(\mathbf{x}) = c] = c, \forall c \in [0, 1]$.

However, models such as neural networks can be miscalibrated. We assume here that $\hat{f}$ is a decision tree classifier and that the extracted feature embedding corresponds to the output of the model $\hat{\pi}$ (i.e. the class probability estimates): $\mathbf{z}_k = \hat{\pi}(\mathbf{x}_k)$ for any target input $\mathbf{x}_k$. For a given target input $\mathbf{x}_k$, $\mathbf{z}_k = \phi(\mathbf{x}_k)$ and $\hat{f}_1(\mathbf{z}_k)$ estimates the probability that the model $\hat{\pi}$ will fail. Therefore, $1 - \hat{f}_1(\mathbf{z}_k)$ is a good candidate to assess the confidence in the model's prediction:

$$1 - \hat{f}_1(\mathbf{z}_k) = 1 - \sum_{i \in \lambda(\mathbf{z}_k)} \frac{\mathcal{L}(\hat{\pi}(\mathbf{x}_i), y_i)}{|\lambda(\mathbf{z}_k)|} = 1 - \sum_{i \in \lambda(\mathbf{z}_k)} \frac{1 - \mathbf{1}_{y_i = \hat{y}_i}}{|\lambda(\mathbf{z}_k)|}, \text{ with } \hat{y}_i = \arg\max_{j \in \mathbb{Y}} \hat{\pi}_j(\mathbf{x}_i) \quad (4)$$

$$1 - \hat{f}_1(\mathbf{z}_k) = 1 - \sum_{i \in \lambda(\mathbf{z}_k)} \frac{1}{|\lambda(\mathbf{z}_k)|} + \sum_{i \in \lambda(\mathbf{z}_k)} \frac{\mathbf{1}_{y_i = \hat{y}_i}}{|\lambda(\mathbf{z}_k)|} = \sum_{i \in \lambda(\hat{\pi}(\mathbf{x}_k))} \frac{\mathbf{1}_{y_i = \hat{y}_i}}{|\lambda(\hat{\pi}(\mathbf{x}_k))|} \quad (5)$$

The right-hand side in (5) is a weighted accuracy given the decision path fulfilled by $\hat{\pi}(\mathbf{x}_k)$. This can be interpreted as confidence recalibration based on the points from $\lambda(\hat{\pi}(\mathbf{x}_k))$; that is, they are in the vicinity of $\mathbf{x}_k$ in terms of probability estimates $\hat{\pi}(\mathbf{x}_k)$ and belong to the same leaf node.

## 3.2 EXPLAINING THE SEVERITY OF SHIFTS

### 3.2.1 EXPLANATION ALGORITHM

We present a sampling-based algorithm that aims to explain why a shift is predicted as potentially harmful for a given prediction, i.e. what feature values contribute to the likelihood of failure assessed by EC. Our method adapts the algorithm from Štrumbelj & Kononenko (2010), which approximates Shapley values by randomly and repeatedly selecting a subset of features instead of all possible coalitions. We make several adaptations to achieve our objective. First, we do not aim to explain the model's predictions; our goal is to justify why a model might fail in a context of distribution shifts. Therefore, we leverage a different kind of value function to estimate the feature contributions. Secondly, the context is multimodal; in particular, we focus on tabular-text data and models. In a nutshell, for a new target input $\mathbf{x}'$ (with $\mathbf{z}' = \phi(\mathbf{x}')$) and Error Classifier $\hat{f}$, we want to understand what contributes to $\hat{f}_1(\mathbf{z}') - \mathbb{E}_{i \sim \mathcal{I}_{val}}[\hat{f}_1(\mathbf{z}_i)]$, in terms of text and tabular feature values.

The approach is described in Algorithm 1 for a target input $\mathbf{x}$, where we compute the average contribution of a tabular feature with index $j$ or a text feature (i.e. token) with index (i.e. position) $j$. We perform $M$ Monte Carlo iterations to approximate the Shapley value. In order to assess the marginal contribution of a feature value with feature index $j$, we construct two new instances $\mathbf{x}_{+j}$ and $\mathbf{x}_{-j}$ from $\mathbf{x}$ by combining the effect of randomness in samples from $\mathcal{S}_{val}$ and in feature indices for tabular and text modalities. As a value function, the Error Classifier $\hat{f}$ is used to assess the contribution of the feature value to the likelihood that $\hat{\pi}$ will fail.

### 3.2.2 MEASURING THE QUALITY OF EXPLANATIONS

To measure the quality of explanations produced by Algorithm 1, we suggest verifying the consistency with outputs generated by other techniques. First, a different value function can be used in Algorithm 1, in order to assess the feature contributions. For instance, deep ensembles (Lakshmi-narayanan et al., 2017) can be leveraged to compute the contribution to uncertainty. In that case, the marginal contribution $\Phi_j^m(\mathbf{x})$ from line 15 in Algorithm 1 equals the difference in predictive entropies computed with $E$ neural networks $p_{\theta_e}$ with parameters $\theta_e$: $\Phi_j^m(\mathbf{x}) = u(\mathbf{z}_{+j}) - u(\mathbf{z}_{-j})$,

$$\text{where } u(\mathbf{z}) = -\sum_{j \in \mathbb{Y}} \left( \frac{1}{E} \sum_{e=1}^{E} p_{\theta_e}(j|\mathbf{z}) \right) \log_2 \left( \frac{1}{E} \sum_{e=1}^{E} p_{\theta_e}(j|\mathbf{z}) \right). \quad (6)$$

Secondly, in Algorithm 1, each perturbation sample ($\mathbf{x}_{+j}$ and $\mathbf{x}_{-j}$) can be modified into a vector $\mathbf{v} \in \{0, 1\}^{(|\mathcal{J}^{tab}| + |\mathcal{J}^{text}|)}$, where each entry from $\mathbf{v}$ equals 1 when the corresponding feature value

---

**Algorithm 1** Explanation algorithm for one feature

---

**Input:** input $\mathbf{x}$ from target dataset $\mathcal{T}$, feature index $j$, index set of tabular features $\mathcal{J}^{tab}$, index set of text features $\mathcal{J}^{text}$, validation source dataset $\mathcal{S}_{val}$, Error Classifier $\hat{f}$, feature encoder $\phi$ component of model $\hat{\pi}$, number of iterations $M$
**Output:** Shapley value $\Phi_j(\mathbf{x})$ for given feature (contribution to probability of error)

1: **for** $m = 1$ to $M$ **do**
2:      Sample $\mathbf{x}^* \sim \mathcal{S}_{val}$
3:      Select random subset of tabular feature indices $\mathcal{R}^{tab} \subset \mathcal{J}^{tab} \backslash \{j\}$
4:      Select random subset of text feature indices $\mathcal{R}^{text} \subset \mathcal{J}^{text} \backslash \{j\}$
5:      Initialize $\mathbf{x}_{+j} \leftarrow \mathbf{x}$                    ▷ here, the subscript is related to features
6:      Replace all the tabular values in $\mathbf{x}_{+j}$ with index in $\mathcal{R}^{tab}$ by corresponding values from $\mathbf{x}^*$
7:      Replace all the text values in $\mathbf{x}_{+j}$ with index in $\mathcal{R}^{text}$ by [MASK] token when these token values are not in $\mathbf{x}^*$
8:      Initialize $\mathbf{x}_{-j} \leftarrow \mathbf{x}_{+j}$
9:      **if** $j \in \mathcal{J}^{tab}$ **then**
10:          Replace the tabular value in $\mathbf{x}_{-j}$ with index $j$ by the corresponding value from $\mathbf{x}^*$
11:      **else**
12:          Replace the text value in $\mathbf{x}_{-j}$ with index $j$ by the [MASK] token when this token value is not in $\mathbf{x}^*$
13:      **end if**
14:      $\mathbf{z}_{+j} \leftarrow \phi(\mathbf{x}_{+j})$ and $\mathbf{z}_{-j} \leftarrow \phi(\mathbf{x}_{-j})$
15:      Compute marginal contribution $\Phi_j^m(\mathbf{x}) \leftarrow \hat{f}_1(\mathbf{z}_{+j}) - \hat{f}_1(\mathbf{z}_{-j})$
16: **end for**

17: Approximated Shapley value $\Phi_j(\mathbf{x}) \leftarrow \dfrac{1}{M} \displaystyle\sum_{m=1}^{M} \Phi_j^m(\mathbf{x})$

---

from $\mathbf{x}$ is present and 0 when it is absent. $|\mathcal{J}^{tab}|$ and $|\mathcal{J}^{text}|$ denote the numbers of tabular features and text tokens, respectively. If we compute Algorithm 1 for the $|\mathcal{J}^{tab}| + |\mathcal{J}^{text}|$ features, we can obtain $2 \times M \times (|\mathcal{J}^{tab}| + |\mathcal{J}^{text}|)$ instances of $\mathbf{v}$ and related $\hat{f}_1(.)$ values (i.e. $\hat{f}_1(\mathbf{z}_{+j})$ and $\hat{f}_1(\mathbf{z}_{-j})$). Then, we can compute the Kernel SHAP weights (Lundberg & Lee, 2017) by fitting a weighted Lasso regression $\hat{r} : \{0, 1\}^{(|\mathcal{J}^{tab}| + |\mathcal{J}^{text}|)} \rightarrow \mathbb{R}$, where $\mathbf{v}$ are the features and $\hat{f}_1(.)$ the response values (or $u(.)$ for deep ensembles). Lastly, the coefficients in this regression function are the Kernel SHAP feature contributions.

The consistency between the outputs obtained with EC and those generated by each of these alternative methods can be assessed, by computing the Pearson correlation coefficients.

## 4 EXPERIMENTS

We empirically test the relevance of our method on various classification datasets. In the appendix, we provide further details on the experimental settings and results (e.g. datasets, data preprocessing, multimodal architectures, baselines, variability in results, ablation studies, computational cost).

### 4.1 SETTINGS

**Datasets.** We test the relevance of our method on 7 classification datasets, with a number of classes ranging from 2 to 100: airbnb, cloth, kick, petfinder, salary, and wine with the 10/100 most frequent classes (referred to as wine10 and wine100, respectively). These datasets have been tested by (Shi et al., 2021) and (Gu & Budhkar, 2021).

**Architectures.** For the multimodal classifier $\hat{\pi}$, we employ four different architectures: (1) AllTextBERT: The tabular features, converted to strings, and the text fields are concatenated and input into BERT-base-uncased (Devlin et al., 2019) as text; (2) LateFuseBERT: A tabular-text dual-stream model with late concatenation of the [CLS] tokens' final hidden states extracted from BERT-base-uncased and a tabular Transformer; (3) AllTextDistilBERT: This architecture is similar to All-

TextBERT, except that we employ DistilBERT-base-uncased (Sanh et al., 2019) instead of BERT; (4) LateFuseDistilBERT: Similar to LateFuseBERT with DistilBERT-base-uncased for the text stream instead of BERT. Each pretrained model is fine-tuned on $\mathcal{S}_{train}$ with a batch size of 32, by minimizing the cross-entropy loss with AdamW algorithm (Loshchilov & Hutter, 2019), with a learning rate of $5e-5$. We use early stopping with patience of 1 for the accuracy on $\mathcal{S}_{val}$. An exponential learning rate scheduler with gamma of 0.9 is employed. We keep the best model in terms of epochs, i.e. with the highest accuracy on $\mathcal{S}_{val}$. Each use case is run over 5 different random dataset partitions.

**Shift type and intensity.** We consider various types of shifts affecting the target dataset $\mathcal{T}$:

- Unimodal covariate shifts: These shifts affect the tabular or text inputs. With *orderSplit*, we sort the data samples of $\mathcal{T}$ by the value of a tabular variable and split the sorted dataset into three sections: $\mathcal{T}_{low}$ (first 5% share), $\mathcal{T}_{mid}$ (5-95%), and $\mathcal{T}_{high}$ (95-100%). The final target dataset is constructed by randomly sampling (with replacement) from $\mathcal{T}_{low} \cup \mathcal{T}_{high}$ and from $\mathcal{T}_{mid}$ with distinct rates. *emptyCategory* randomly replaces the values of categorical variables with empty values. With *typos*, we insert random typographical errors into the text field (e.g. swapping, removing, adding, or replacing characters; adding or removing spaces). With *seqLengthSplit*, we sort the data samples of $\mathcal{T}$ in ascending or descending order based on the text field length. We then split the sorted dataset into two sections: $\mathcal{T}_1$ (first 10% share) and $\mathcal{T}_2$ (90%). The final target dataset is constructed by randomly sampling (with replacement) from each section with distinct rates. *cutText* randomly truncates the text field by removing a part of the end of the text. Lastly, *abbrev* randomly replaces words by abbreviations provided by a given list (e.g. *especially* becomes *esp*).

- Multimodal covariate shifts. This is achieved through the combination of shifts affecting the inputs of both modalities: *orderSplit-typos, emptyCategory-typos, orderSplit-seqLengthSplit, orderSplit-cutText, orderSplit-abbrev*.

- Out-of-domain: With *newClass*, $\mathcal{T}$ includes a proportion of samples where the true label is not one of the $C$ classes. For cloth, pet, and salary, the source dataset is constructed after removing the samples from the minority class, which are then randomly inserted into $\mathcal{T}$. For wine10 and wine100, we insert into $\mathcal{T}$ samples from unknown classes originating from wine100 and wine200, respectively.

Further, we implement 3 degrees of shift intensity corresponding to various levels of sampling rates (*orderSplit, seqLengthSplit, newClass*), distinct percentages of affected target data rows (*emptyCategory, abbrev, cutText*), various numbers of shifts (*typos*), or different proportions of text to remove (*cutText*). Lastly, as a reference, we also test the methods on the unchanged target dataset (*noShift*).

**Evaluation.** For each experiment, all the methods are calibrated on the validation source data $\mathcal{S}_{val}$ and evaluated on the same target dataset $\mathcal{T}$ with a size of 1000 rows. The final hidden state of the classification token [CLS] (referred to as $\mathbf{z}_{[CLS]}^{last}$) and the softmax output $\hat{\pi}(\mathbf{x})$ are extracted from $\hat{\pi}$. For LateFuseBERT, $\mathbf{z}_{[CLS]}^{last}$ is the concatenation of the text and tabular Transformers' final hidden states for the [CLS] tokens (i.e. states before the classification head). For EC, we use a random forest algorithm with the default hyperparameter setting from *Scikit-learn* Python package (Pedregosa et al., 2011). $\mathbf{z}_{[CLS]}^{last}$ and $\hat{\pi}(\mathbf{x})$ are concatenated and used as features for EC. Our method is compared to the following baselines previously described in section 2.

15 baselines are used for unsupervised error rate estimation, where the scores are computed for a given target dataset $\mathcal{T}$: (1) JSD: The Jensen-Shannon Distance between the (validation) source and target distributions of maximum confidences (i.e. maximum softmax probabilities); (2) AC: One minus the average maximum confidence over the target dataset; (3) ACSC: One minus the average maximum confidence after applying temperature scaling to the softmax output; (4) MMD: maximum mean discrepancy between the source and target samples of $\mathbf{z}_{[CLS]}^{last}$; (5) DOC: The error rate estimated with the Difference Of Confidences between source and target; (6) ATC: The error rate estimated with the Average Thresholded Confidence; (7) MAND: The error rate estimated with Mandoline; (8) MCD: With Monte Carlo dropout, the uncertainty is assessed with the average of the predictive entropies over $\mathcal{T}$, computed after performing 5 forward passes of $\hat{\pi}$; (9) DC: The AUROC metric of the domain classifier trained with $\mathbf{z}_{[CLS]}^{last}$ as features; (10) CP: The mean prediction set size computed with the weighted conformal prediction (Tibshirani et al., 2019) based on LAC method (Sadinle et al., 2019); (11) DNN: The average distance to the $k$-th neighbor ($k = 10$) from the source data

with the deep nearest neighbors fitted with $\mathbf{z}_{[\mathrm{CLS}]}^{\mathrm{last}}$ as features; (12) ENRG: The mean energy score computed over the target dataset; (13) TCP: One minus the average true class probability estimated with a neural network trained with $\mathbf{z}_{[\mathrm{CLS}]}^{\mathrm{last}}$; (14) DENS: The uncertainty is assessed with the average of the predictive entropies over $\mathcal{T}$, after averaging the probabilities from a deep ensemble of 5 neural networks trained with $\mathbf{z}_{[\mathrm{CLS}]}^{\mathrm{last}}$; (15) PNORM: The distance as the sum of squared differences between the original and new model's parameters, after training each model based on $\mathbf{z}_{[\mathrm{CLS}]}^{\mathrm{last}}$.

Some methods can exclusively be assessed for OOD performance estimation (e.g. JSD, MMD). Therefore, only 9 baselines are used for the error detection task, where the scores are computed for a given target input: (1) AC: One minus the maximum confidence for a given target input; (2) AC-scaled: One minus the maximum confidence after applying temperature scaling to the softmax output; (3) MCD: With Monte Carlo dropout, the uncertainty is assessed with the predictive entropy, after computing 5 forward passes of $\hat{\pi}$; (4) DC: The class 1's predicted probability; (5) CP: The prediction set size computed with the weighted conformal prediction based on LAC method; (6) DNN: The distance to the $k$-th neighbor ($k = 10$); (7) ENRG: The energy score; (8) TCP: One minus the true class probability; (9) DENS: The uncertainty is assessed with the predictive entropy.

For a given architecture (e.g. LateFuseBERT), we evaluate how each method performs on unsupervised error rate estimation by computing the Spearman's rank correlation $\rho$ between the scores and the actual error rates on the target dataset, over different random dataset partitions (seeds), shift types and intensities. The performance on error detection is assessed by computing AUROC with all the target data from different seeds, shift types and intensities: we calculate the scores for accurate (label 0) and incorrect (label 1) predictions, and quantify how well these two labels are separated for a range of thresholds. Lastly, we also perform ablation studies to compare the results of EC with (1) Ablation 1: an Error Classifier using only $\mathbf{z}_{[\mathrm{CLS}]}^{\mathrm{last}}$ as features, or (2) Ablation 2: EC leveraging only the classifier's output $\hat{\pi}(\mathbf{x})$. The results of the ablation studies are in appendix I.

**Explanation algorithm.** We experiment with two different value functions: the Error Classifier and deep ensembles. In order to accelerate the computation of Shapley values, we stop the iterations when a convergence criteria is reached. To achieve that, we first compute the maximum absolute difference between the previous and updated Shapley values, every 10 iterations and for each value function. We end the process when the maximum of these two values is lower than $0.01$.

## 4.2 RESULTS

Table 1: **Evaluation of the methods for LateFuseBERT**, computed on the target data for 5 random seeds, and different shift types and intensities. Error rate estimation is assessed with the Spearman's rank correlation ($\rho$). Error detection is evaluated with AUROC (auc) and is only applicable to EC and 9 baselines. For a given dataset and task, the best result is in **bold** (higher is better). The variability in results is displayed in appendix H.

| Method | airbnb | | cloth | | kick | | petfinder | | salary | | wine10 | | wine100 | |
|---|---|---|---|---|---|---|---|---|---|---|---|---|---|---|
| | $\rho$ | auc | $\rho$ | auc | $\rho$ | auc | $\rho$ | auc | $\rho$ | auc | $\rho$ | auc | $\rho$ | auc |
| AC | 0.412 | 0.628 | 0.366 | 0.744 | 0.582 | 0.842 | 0.102 | 0.586 | 0.190 | 0.639 | 0.774 | 0.840 | 0.767 | **0.858** |
| ACSC | 0.366 | **0.638** | 0.375 | 0.710 | 0.688 | 0.842 | 0.109 | 0.589 | 0.417 | **0.648** | 0.826 | 0.841 | 0.302 | 0.649 |
| ATC | 0.258 | | 0.460 | | 0.728 | | 0.427 | | 0.459 | | 0.864 | | 0.906 | |
| CP | 0.030 | 0.625 | 0.518 | 0.705 | 0.676 | 0.615 | 0.322 | 0.573 | 0.307 | 0.623 | 0.751 | 0.775 | 0.784 | 0.849 |
| DC | 0.268 | 0.511 | 0.429 | 0.543 | 0.048 | 0.489 | 0.016 | 0.502 | 0.154 | 0.489 | 0.590 | 0.609 | 0.404 | 0.586 |
| DENS | **0.455** | 0.613 | 0.584 | **0.759** | 0.526 | 0.863 | 0.353 | 0.567 | 0.213 | 0.646 | 0.797 | 0.840 | 0.898 | 0.843 |
| DNN | -0.200 | 0.532 | 0.166 | 0.665 | -0.233 | 0.575 | -0.293 | 0.473 | 0.141 | 0.530 | 0.429 | 0.736 | 0.531 | 0.753 |
| DOC | 0.289 | | 0.647 | | 0.718 | | 0.554 | | **0.629** | | 0.910 | | 0.916 | |
| EC | 0.203 | 0.632 | **0.804** | 0.755 | **0.835** | **0.885** | 0.523 | **0.608** | 0.450 | 0.622 | 0.904 | **0.843** | **0.927** | 0.857 |
| ENRG | 0.359 | 0.620 | 0.218 | 0.673 | 0.485 | 0.651 | -0.000 | 0.508 | -0.038 | 0.561 | 0.610 | 0.793 | 0.675 | 0.841 |
| JSD | 0.124 | | 0.493 | | -0.101 | | -0.112 | | 0.170 | | 0.682 | | 0.497 | |
| MAND | 0.279 | | 0.634 | | 0.736 | | **0.560** | | 0.600 | | **0.911** | | 0.916 | |
| MCD | 0.371 | **0.638** | 0.452 | 0.748 | 0.625 | 0.842 | 0.107 | 0.578 | 0.164 | 0.638 | 0.791 | 0.830 | 0.760 | 0.850 |
| MMD | 0.307 | | 0.526 | | -0.150 | | -0.122 | | 0.200 | | 0.647 | | 0.459 | |
| PNORM | -0.227 | | 0.009 | | -0.394 | | -0.311 | | 0.168 | | 0.133 | | 0.354 | |
| TCP | -0.032 | 0.533 | 0.240 | 0.560 | 0.235 | 0.585 | 0.522 | 0.528 | 0.008 | 0.531 | -0.002 | 0.556 | 0.174 | 0.562 |

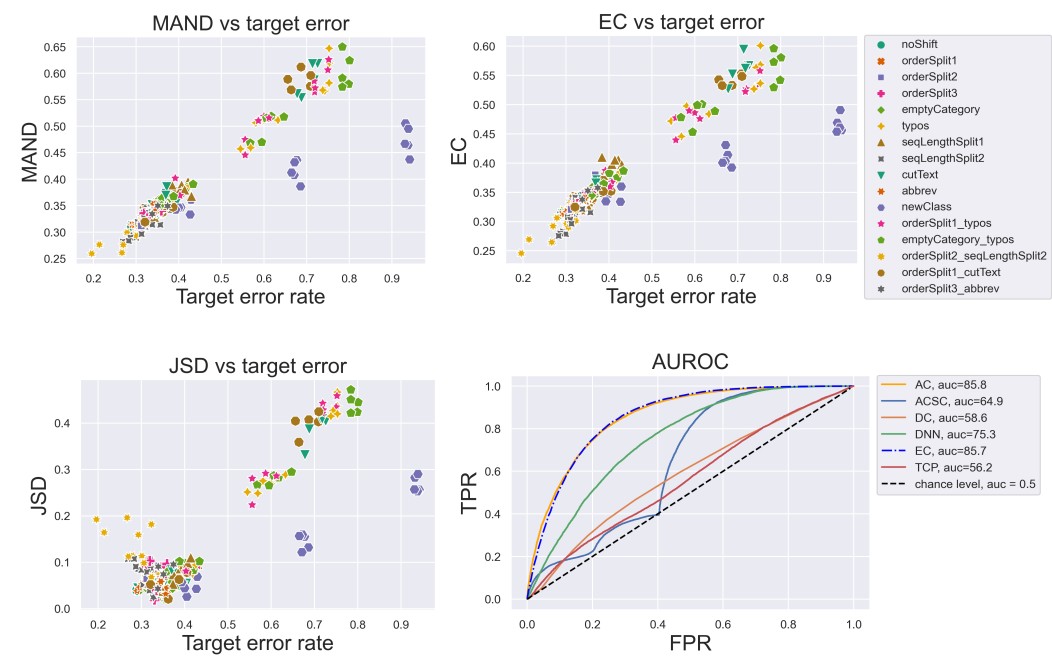

Figure 2: From top to bottom , left to right. **Score versus true error rate** for the task of unsupervised performance estimation, for MAND, EC, and JSD, on wine100 target data (with LateFuseBERT), by shift type of various intensities and different seeds. *orderSplit1/2/3* correspond to 3 different tabular features affected by the shift, while *seqLengthSplit1/2* correspond to ascending/descending order, respectively. **AUROC curves** for a few methods on error detection task, computed on wine100 target data (with LateFuseBERT), across seeds, shift types, and intensities.

**Evaluation of the methods.** With LateFuseBERT architecture, Table 1 shows that no method significantly outperforms the other ones across different datasets, shift types, and intensities. However, EC achieves a strong performance on both tasks: unsupervised performance estimation (first rank in 3 use cases) and error detection (first rank in 3 use cases, second rank in 2 use cases). The first two scatter plots from Figure 2 compare EC with the best baseline for unsupervised performance estimation (MAND) on wine100 target data. Both methods perform well for most of the shifts, except *newClass*. The severity of this out-of-domain shift seems to be more difficult to assess. The third scatter plot (bottom left) shows that JSD is less appropriate for this use case, as the monotony is less obvious. The last plot from Figure 2 displays the AUROC curves computed on wine100 for a few methods assessed on the error detection task. EC almost matches AC which is the best method in that case. Table 2 for AllTextBERT (and Tables 4 / 5 in appendix G for LateFuseDistilBERT / AllTextDistilBERT) confirm that EC achieves solid performance across the various architectures and tasks. Regarding unsupervised performance estimation, the methods that are specialized in this task (esp. MAND and DOC) tend to achieve better results than the baselines that produce scores for both tasks.

**Explanation algorithm.** Figure 3 (left) shows an example from wine10 where the shift type is *newClass*; that is, the instance is from an unknown class. The goal is to predict the variety of grapes. The plot displays the outputs from the explanation algorithm computed with EC as value function. The probability of error assessed by EC is $77\%$, which is significantly higher than the mean probability of error on $\mathcal{S}_{val}$ ($19\%$). Further, the top 10 feature contributions to the probability of error are all positive and related to tabular and text features, which evidence that the input is uncommon. Figure 3 (right) shows the Pearson correlation matrices between the feature contributions computed with different value functions (EC, DENS) and algorithms (explanation algorithm 1, Kernel SHAP). The first correlation matrix is related to the shifted instance from Figure 1 (right) and indicates consistency between all the explanation methods. On the other hand, the second correlation matrix corresponds to the example from Figure 3 (left). Only the outputs from Algorithm 1 and Kernel

SHAP computed with EC are consistent, which may be more reliable than the other methods in that specific case. Other examples are included in appendix K.

Table 2: **Evaluation of the methods for AllTextBERT**, computed on target data for 5 random seeds, and different shift types and intensities. Error rate estimation is assessed with the Spearman's rank correlation ($\rho$). Error detection is evaluated with AUROC (auc) and is only applicable to EC and 9 baselines. The variability in results is displayed in appendix H.

| Method | airbnb $\rho$ | airbnb auc | cloth $\rho$ | cloth auc | kick $\rho$ | kick auc | petfinder $\rho$ | petfinder auc | salary $\rho$ | salary auc | wine10 $\rho$ | wine10 auc | wine100 $\rho$ | wine100 auc |
|---|---|---|---|---|---|---|---|---|---|---|---|---|---|---|
| AC | 0.092 | 0.603 | 0.293 | 0.760 | 0.350 | 0.872 | 0.155 | 0.553 | -0.011 | 0.643 | 0.733 | 0.837 | 0.783 | **0.853** |
| ACSC | 0.195 | 0.612 | 0.366 | **0.764** | 0.342 | 0.872 | 0.159 | 0.559 | -0.014 | 0.640 | 0.466 | 0.649 | 0.484 | 0.734 |
| ATC | 0.214 | | 0.538 | | **0.772** | | 0.376 | | **0.305** | | 0.885 | | 0.881 | |
| CP | 0.295 | 0.611 | 0.584 | 0.711 | 0.551 | 0.591 | 0.319 | 0.557 | 0.249 | 0.618 | 0.846 | 0.756 | 0.744 | 0.839 |
| DC | 0.299 | 0.510 | 0.503 | 0.543 | -0.221 | 0.468 | 0.253 | 0.503 | 0.169 | 0.502 | 0.604 | 0.619 | 0.416 | 0.571 |
| DENS | 0.124 | 0.609 | 0.662 | 0.761 | 0.555 | **0.884** | -0.040 | 0.539 | 0.257 | **0.649** | 0.834 | 0.828 | 0.794 | 0.827 |
| DNN | -0.080 | 0.527 | 0.268 | 0.643 | 0.025 | 0.493 | 0.034 | 0.506 | 0.264 | 0.546 | 0.476 | 0.764 | 0.625 | 0.699 |
| DOC | 0.398 | | 0.749 | | 0.760 | | **0.393** | | 0.292 | | 0.906 | | 0.884 | |
| EC (ours) | **0.425** | **0.633** | **0.789** | 0.756 | 0.759 | 0.877 | 0.360 | **0.571** | 0.236 | 0.615 | 0.903 | **0.849** | **0.907** | 0.847 |
| ENRG | 0.152 | 0.596 | 0.292 | 0.738 | 0.323 | 0.759 | 0.064 | 0.505 | -0.000 | 0.580 | 0.553 | 0.792 | 0.764 | 0.842 |
| JSD | 0.214 | | 0.656 | | -0.465 | | 0.290 | | 0.152 | | 0.646 | | 0.430 | |
| MAND | 0.405 | | 0.746 | | 0.761 | | 0.368 | | 0.297 | | **0.908** | | 0.884 | |
| MCD | 0.154 | 0.600 | 0.372 | 0.758 | 0.320 | 0.873 | 0.037 | 0.547 | 0.013 | 0.632 | 0.709 | 0.827 | 0.767 | 0.844 |
| MMD | 0.233 | | 0.608 | | -0.390 | | 0.277 | | 0.247 | | 0.673 | | 0.451 | |
| PNORM | -0.160 | | 0.288 | | 0.429 | | 0.169 | | 0.291 | | 0.088 | | 0.349 | |
| TCP | 0.388 | 0.519 | 0.050 | 0.546 | -0.043 | 0.610 | 0.009 | 0.510 | 0.085 | 0.517 | 0.213 | 0.557 | 0.257 | 0.586 |

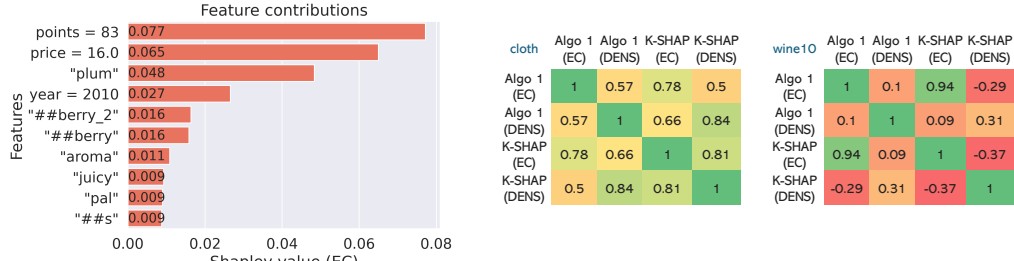

Figure 3: *Left*: **Top 10 feature contributions** computed with Algorithm 1 (EC as value function) for an out-of-domain instance, where the classifier is LateFuseBERT and the dataset is wine10. The text is "Candied cherry and raspberry aromas are dilute by Mendoza standards. The palate is regular at best, with juicy generic flavors of plum and raspberry (...)." The tabular variables are country ("Argentina"), year (2010), points (83), price (16.0). EC value: 77%. *Right*: **Pearson correlation** matrices between the outputs of various explanations methods: value functions (EC, DENS), and algorithms (Algo 1: Algorithm 1, K-SHAP: Kernel SHAP). The first matrix is related to the example from Figure 1 (right), whereas the second one is related to the example from Figure 3 (left).

## 5 CONCLUSION

We introduced a method to compute and explain the likelihood of failure in classification tasks and in OOD contexts. We compared our method to 15 baselines and evidenced that the Error Classifier can be a useful approach for estimating performance and detecting errors on unlabeled data. The outputs of the explanation algorithm proposed in this paper can be relevant to locally understand the source of distribution shifts. The quality of explanations can be assessed by using various value functions (ex. DENS vs EC) or algorithms (e.g. Kernel SHAP). These results are specific to the shifts and the classification models tested here. Therefore, it would be useful to experiment with different settings. Lastly, future work could also address the case of other modalities (e.g. image).

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

# A    APPENDIX: TABLE OF CONTENTS

## B  DATASETS AND SAMPLING

All the datasets are publicly available with one of these licenses: "CC0: Public Domain", "Competition Data", or "CC BY-NC-SA 4.0". These datasets can be accessed and used for the purpose of academic research. The text fields are in English.

In Table 3, we give more details on the datasets:

- airbnb[1]: the task is to predict the price range of Airbnb listings. The text fields are listing descriptions.

- cloth[2]: the goal is to classify the sentiment (represented as a class) of user reviews regarding clothing items. The text fields are customer reviews.

- kick[3]: the task is to predict whether a proposed project will achieve its funding goal. The text fields are project descriptions.

- petfinder[4]: the goal is to predict the speed range at which a pet is adopted. The text fields are profile write-ups for the pets.

- salary[5]: the task is to predict the salary range based on data scientist job postings. The text fields are job descriptions.

- wine[6]: the goal is to predict the variety of grapes. The text fields are wine tasting descriptions.

For some of the use cases, we employ the original training dataset as the test dataset does not include the true labels (competition data). In that case, we consider the training dataset as the modeling data which is then randomly split into training-validation-target subsets. The datasets are partitioned as follows: (1) The initial dataset is randomly split into two disjoint temporary (80% share) and target $\mathcal{T}$ (20% share) subsets, respectively; (2) The temporary dataset is randomly split into two disjoint training $\mathcal{S}_{train}$ (80% share) and validation $\mathcal{S}_{val}$ (20% share) subsets, respectively. For the evaluation of the methods, 1000 rows are randomly extracted from the original target dataset.

Table 3: Information on datasets: number of samples in training dataset, number of numerical/categorical features, number of classes (before removing classes for the *newClass* shift).

| Dataset | # Train | # Num | # Cat | # Class |
|---|---|---|---|---|
| **airbnb** | 4,372 | 27 | 23 | 10 |
| **cloth** | 13,955 | 2 | 3 | 5 |
| **kick** | 69,194 | 3 | 3 | 2 |
| **petfinder** | 9,324 | 5 | 14 | 5 |
| **salary** | 10,975 | 1 | 2 | 6 |
| **wine10** | 39,320 | 2 | 2 | 10 |
| **wine100** | 65,398 | 2 | 2 | 100 |

## C  DATA PREPROCESSING

**Feature engineering.** When the dataset contains several text fields, these are concatenated in order to obtain a single field. Rows with missing values are dropped and duplicate rows removed. The list of final features for each dataset is described below. We also mention here additional features that were created from the raw dataset.

---

[1] https://www.kaggle.com/datasets/tylerx/melbourne-airbnb-open-data
[2] https://www.kaggle.com/datasets/nicapotato/womens-ecommerce-clothing-reviews
[3] https://www.kaggle.com/datasets/codename007/funding-successful-projects
[4] https://www.kaggle.com/competitions/petfinder-adoption-prediction/data
[5] https://machinehack.com/hackathons/predict_the_data_scientists_salary_in_india_hackathon/overview
[6] https://www.kaggle.com/datasets/zynicide/wine-reviews

- airbnb: for this dataset only, we discretize the target variable by employing quantile binning (ten intervals with equal share of data). We also create two new features *host_since_year* and *last_review_year* by extracting the year from *host_since* and *last_review* respectively. Categorical variables: *host_location, host_since_year, host_is_superhost, host_neighborhood, host_has_profile_pic, host_identity_verified, neighborhood, city, smart_location, suburb, state, is_location_exact, property_type, room_type, bed_type, instant_bookable, cancellation_policy, require_guest_profile_picture, require_guest_phone_verification, host_response_time, calendar_updated, host_verifications, last_review_year*; numerical variables: *host_response_rate, latitude, longitude, accommodates, bathrooms, bedrooms, beds, security_deposit, cleaning_fee, guests_included, extra_people, minimum_nights, maximum_nights, availability_30, availability_60, availability_90, availability_365, number_of_reviews, review_scores_rating, review_scores_accuracy, review_scores_cleanliness, review_scores_checkin, review_scores_communication, review_scores_location, review_scores_value, calculated_host_listings_count, reviews_per_month*; text fields: *name, summary, description*.

- cloth: categorical variables: *Division Name, Department Name, Class Name*; numerical variables: *Age, Positive Feedback Count*; text fields: *Title, Review Text*.

- kick: we compute the duration to launch (in days) with *deadline* and *launched_at*. We also log-transform *goal*. Categorical variables: *country, currency, disable_communication*; numerical variables: *log_goal, backers_count, duration*; text fields: *name, desc*.

- petfinder: Categorical variables: *Type, Breed1, Breed2, Gender, Color1, Color2, Color3, MaturitySize, FurLength, Vaccinated, Dewormed, Sterilized, Health, State*; numerical variables: *Age, Quantity, Fee, VideoAmt, PhotoAmt*; text field: *Description*.

- salary: Categorical variables: *location, company_name_encoded*; numerical variables: *experience_int*; text fields: *job_description, job_desig, key_skills*.

- wine10 and wine100: we extract the *year* from *title*. Categorical variables: *country, year*; numerical variables: *points, price*; text field: *description*.

**Text preprocessing.** We perform the following text preprocessing: we keep words, numbers, and whitespaces. We then use the BERT-base-uncased or DistilBERT-base-uncased tokenizer based on WordPiece. For the text sequence length, the value is set to the 0.9 quantile of the text field lengths' distribution in the source dataset. We then take the minimum of this latter value and $512$ as this is the maximum sequence length for BERT models. We use truncation and padding to the fixed maximum length.

**Attention mask.** We use key attention masks in order to specify which text tokens should be ignored (i.e. "padding") for the purpose of attention.

**Class preprocessing.** In order to implement out-of-domain shifts, some data points of specific labels are removed. For cloth, pet, and salary, the training dataset is constructed after removing the samples from the minority class, which are then randomly inserted into $\mathcal{T}$. For wine10 and wine100, we insert into $\mathcal{T}$ samples from unknown classes originating from wine100 and wine200, respectively.

## D  CLASSIFICATION MODEL ARCHITECTURES

**LateFuse architecture.** The architecture is detailed in Figure 4 (right) with BERT-base-uncased for the text stream. For numerical features, we first perform standard scaling. Embeddings of the LateFuse architecture are constructed with linear functions. A linear function applies the following transformation to a scalar feature value $x \in \mathbb{R}$: $x.W_{num} + b$ where $W_{num} \in \mathbb{R}^d$ and the bias $b \in \mathbb{R}^d$. For categorical features, we encode them as category embeddings. In that latter case, the corresponding embedding is computed as $e^T W_{cat}$ where $e \in \mathbb{R}^{n_c \times 1}$ is a one-hot-vector for the associated categorical feature, $n_c$ denote the number of categories for this feature, and $W_{cat} \in \mathbb{R}^{n_c \times d}$. A classification token [CLS] is then added to the beginning of the tabular embedding sequence. The tabular Transformer with self-attention has the following architecture: 3 layers, 8 attention heads, feed-forward dimension of 768, embedding dimension of 768. The dropout (rate 0.1) is applied to the category embeddings, the tabular Transformer (attention, feed-forward networks), and the final

fully-connected networks. The text and tabular Transformer's final hidden states of the [CLS] tokens are concatenated before being projected through fully-connected layers to produce the logits. The uniform weight initialization for the category/linear embeddings and the final fully-connected networks is based on Kaiming (He et al., 2015). The final fully-connected layers can be described as follows: $\text{FC}(x) = \text{Linear}(\text{Dropout}(\text{ReLU}(\text{Linear}(x))))$ where the output has a dimension of $C$ (number of classes).

**AllText architecture.** The architecture is detailed in Figure 4 (left) with BERT-base-uncased. The tabular features, converted to strings, and the text fields are concatenated and input into BERT-base-uncased as text. The final hidden state of the [CLS] token (i.e. before the classification head) are projected through fully-connected layers to produce the logits. The uniform weight initialization for the final fully-connected networks is based on Kaiming. The final fully-connected layers can be described as follows: $\text{FC}(x) = \text{Linear}(\text{Dropout}(\text{ReLU}(\text{Linear}(x))))$ where the output has a dimension of $C$ (number of classes). The dropout rate is 0.1 in the final fully-connected networks.

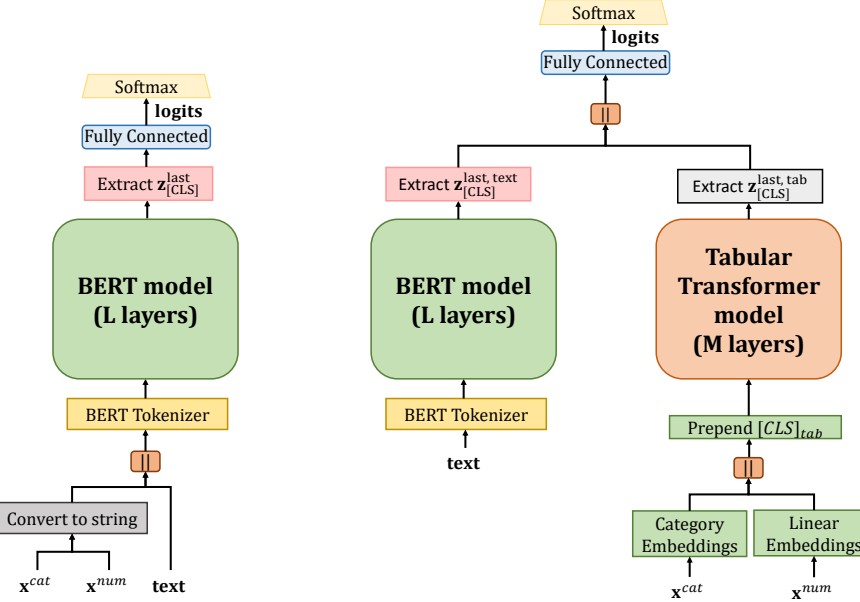

Figure 4: **Classification model architectures**. *Left:* AllTextBERT architecture. *Right:* LateFuse-BERT architecture.

# E   SHIFT GENERATION

We provide further details on the shift generation techniques described in Section 4:

- *orderSplit*: The three intensities of shift correspond to different levels of sampling rates used to construct the final target dataset: We sample with a rate of 10%, 50%, or 90% from $\mathcal{T}_{low} \cup \mathcal{T}_{high}$, respectively. This shift is applied to a feature selected randomly: e.g. *orderSplit1*, *orderSplit2*, *orderSplit3* correspond to 3 different seeds used to select a given variable from the original feature pool.

- *emptyCategory*: The three intensities correspond to various percentages of target data affected by this shift: 10%, 50%, or 90%.

- *typos*: We use the Python package typo (https://github.com/ranvijaykumar/typo) in order to generate typographical errors. The three intensities correspond to different numbers of typos affecting the text field of each target observation: 5, 25, or 50. It is worth noting that, for AllTextBERT, the typos are applied to the original text fields, i.e. before concatenating tabular features (converted to strings) and text fields.

- *seqLengthSplit*: The three intensities of shift correspond to different levels of sampling rates used to construct the final target dataset: We sample with a rate of 10%, 50%, or 90% from $\mathcal{T}_1$, respectively. $\mathcal{T}_1$ includes either the 10% shortest (scenario 1) or longest (scenario 2) text fields.

- *cutText*: The three intensities correspond to various proportions of target data affected by this shift (10%, 50%, or 90%) and different percentage of text to truncate (10%, 50%, or 90%) from the original text field.

- *abbrev*: The three intensities correspond to various percentages of target data affected by this shift : 10%, 50%, or 90%. The abbreviation dataset includes a list of roughly 2000 abbreviations from Oxford English Dictionary (`https://www.oed.com/information/understanding-entries/abbreviations/`).

- *orderSplit-typos, emptyCategory-typos, orderSplit-seqLengthSplit, orderSplit-cutText, orderSplit-abbrev*: These multimodal shifts are just combinations of unimodal shifts, and follow the same rules as previously described.

- *newClass*: The three intensities of shift correspond to different levels of sampling rates used to construct the final target dataset: We sample with respective rates of 10%, 50%, or 90% from the new dataset with unknown classes (minority class for cloth, pet, and salary; unknown classes from wine100 for wine 10; unknown classes from wine200 for wine 100). wine200 is constructed similarly to wine10 or wine100, it includes the 200 most frequent classes from wine dataset.

## F  SHIFT EVALUATION: DETAILS ON BASELINES

**[CLS] token's final hidden state as feature.** For the methods leveraging the [CLS] tokens' final hidden states, it is worth mentioning that when $\hat{\pi}$ is based on LateFuse architecture, the text and tabular hidden states are concatenated (see Figure 4). In that case, the final vector is of dimension $2 \times 768$.

**Further details on baselines.** We provide further details for some of the methods used to assess shift severity:

- JSD: The number of bins used to discretize the maximum softmax probability distributions is set to 10.

- ACSC: Temperature scaling is performed on the source validation dataset ($\mathcal{S}_{val}$). The temperature is set by optimizing the Expected Calibration Error (ECE) with the L-BFGS algorithm.

- MMD: The computation is based on the Radial basis function kernel (RBF).

- MAND: The error rate is estimated based on the 0-1 loss error importance re-weighting with one slice based on the classification model's maximum confidence.

- MCD: We enable the dropout layers from $\hat{\pi}$ during test-time. The dropout probability is set to 0.1. For each target example, we perform $P = 5$ forwards passes with $\hat{\pi}$ and corresponding parameters $\theta_p$. Then, we calculate the total uncertainty (entropy) after averaging the predicted probabilities:

$$u(\mathbf{x}) = -\sum_{j \in \mathbb{Y}} \left( \frac{1}{P} \sum_{p=1}^{P} \hat{\pi}_j(\mathbf{x}; \theta_p) \right) \log_2 \left( \frac{1}{P} \sum_{p=1}^{P} \hat{\pi}_j(\mathbf{x}; \theta_p) \right) \tag{7}$$

- DC: For the domain classifier, we employ a Random Forest with 10 estimators. We divide both the source data and target data into two halves, using the first half to train a domain classifier to classify source (class 0) and target (class 1) data. We then apply this model to the second half and compute the AUROC. We follow the same process by selecting the second half to fit the domain classifier and computing the AUROC on the first half. Lastly, we average the 2 AUROC values.

- CP: For the weighted conformal prediction, we compute weighted quantiles. Each weight is computed with the domain classifier as $\hat{p}_{dc}(\mathbf{z})/(1 - \hat{p}_{dc}(\mathbf{z}))$, where $\hat{p}_{dc}(\mathbf{z})$ is the probability that the input is from the target given $\mathbf{z}$. This approach is suggested in (Tibshirani et al., 2019). With the LAC method, the conformity score corresponds to one minus the probability of the true class. For this baseline, we set the quantile to 90%, which is the expected coverage.

- DNN: The feature space is normalized with the l2 norm as a pre-requisite, as advised in (Sun et al., 2022).

- ENRG: In the energy score formula, we set the temperature to 1.

- TCP: The neural network used to estimate the true class probability has the following architecture: $\text{NN}(x) = \text{Linear}(\text{Dropout}(\text{ReLU}(\text{Linear}(x))))$. The dropout probability is set to $0.1$. The (input shape, output shape) for the first linear layer is compatible with the dimension of $\mathbf{z}$ ($768 \times 768$ for the AllText architecture and $(2 \times 768) \times (2 \times 768)$ for LateFuse architecture). As this is a regression task, the final output has dimension 1. The mean squared error loss is optimized with Adam (learning rate of $1e-3$) for 10 epochs and batch size of 32.

- DENS: An ensemble of 5 neural networks is trained, where each neural network has the following architecture: $\text{NN}(x) = \text{Linear}(\text{Dropout}(\text{ReLU}(\text{Linear}(x))))$. The dropout probability is set to 0.1. The (input shape, output shape) for the first linear layer is compatible with the dimension of $\mathbf{z}$ ($768 \times 768$ for the AllText architecture and $(2 \times 768) \times (2 \times 768)$ for LateFuse architecture). As this is a classification task, the final output has dimension $C$. The cross-entropy loss is optimized with Adam (learning rate of $1e-3$) for 10 epochs and batch size of 32. For each target example, we compute the total uncertainty (predictive entropy), after averaging the predicted probabilities generated by $E = 5$ neural networks $p_{\theta_e}$ with parameters $\theta_e$:

$$u(\mathbf{z}) = -\sum_{j \in \mathbb{Y}} \left( \frac{1}{E} \sum_{e=1}^{E} p_{\theta_e}(j|\mathbf{z}) \right) \log_2 \left( \frac{1}{E} \sum_{e=1}^{E} p_{\theta_e}(j|\mathbf{z}) \right) \tag{8}$$

- PNORM: The in-distribution model and the new model have the same architecture: $\text{NN}(x) = \text{Linear}(\text{Dropout}(\text{ReLU}(\text{Linear}(x))))$. The dropout probability is set to 0.1. The (input shape, output shape) for the first linear layer is compatible with the dimension of $\mathbf{z}$ ($768 \times 768$ for the AllText architecture and $(2 \times 768) \times (2 \times 768)$ for LateFuse architecture). As this is a classification task, the final output has dimension $C$. The cross-entropy loss is optimized with Adam (learning rate of $1e-3$) for 10 epochs and batch size of 32.

## G  RESULTS FOR DISTILBERT

The results for LateFuseDistilBERT architecture are presented in Table 4. The results for All-TextDistilBERT architecture are presented in Table 5.

Table 4: **Evaluation of the methods for LateFuseDistilBERT**, computed on target data for 5 random seeds, and different shift types and intensities. Error rate estimation is assessed with the Spearman's rank correlation ($\rho$). Error detection is evaluated with AUROC (auc) and is only applicable to EC and 9 baselines. For a given dataset and task, the best result is in **bold** (higher is better). The variability in results is displayed in appendix.

| Method | airbnb $\rho$ | auc | cloth $\rho$ | auc | kick $\rho$ | auc | petfinder $\rho$ | auc | salary $\rho$ | auc | wine10 $\rho$ | auc | wine100 $\rho$ | auc |
|---|---|---|---|---|---|---|---|---|---|---|---|---|---|---|
| AC | **0.643** | 0.620 | 0.306 | 0.744 | 0.397 | 0.850 | 0.073 | 0.572 | 0.264 | 0.636 | 0.635 | 0.828 | 0.736 | 0.857 |
| ACSC | 0.628 | 0.632 | 0.332 | 0.751 | 0.591 | 0.850 | 0.114 | 0.572 | **0.409** | 0.642 | 0.643 | 0.726 | 0.818 | **0.860** |
| ATC | 0.501 | | 0.541 | | 0.600 | | 0.187 | | 0.272 | | 0.841 | | 0.876 | |
| CP | 0.408 | 0.633 | **0.821** | 0.725 | 0.590 | 0.602 | 0.289 | 0.555 | 0.344 | 0.617 | 0.733 | 0.766 | 0.754 | 0.847 |
| DC | 0.189 | 0.512 | 0.404 | 0.547 | 0.013 | 0.495 | 0.062 | 0.503 | 0.131 | 0.487 | 0.604 | 0.620 | 0.438 | 0.588 |
| DENS | 0.604 | 0.614 | 0.644 | **0.764** | 0.300 | 0.862 | 0.096 | 0.568 | 0.171 | **0.643** | 0.772 | 0.836 | 0.808 | 0.837 |
| DNN | -0.562 | 0.505 | 0.237 | 0.649 | -0.030 | 0.631 | 0.068 | 0.497 | 0.198 | 0.545 | 0.429 | 0.747 | 0.585 | 0.746 |
| DOC | 0.497 | | 0.763 | | 0.621 | | 0.371 | | 0.398 | | 0.841 | | 0.902 | |
| EC (ours) | 0.468 | **0.636** | 0.791 | 0.761 | **0.718** | **0.879** | 0.390 | **0.597** | 0.366 | 0.618 | 0.830 | **0.836** | **0.912** | 0.849 |
| ENRG | 0.633 | 0.611 | 0.142 | 0.663 | 0.320 | 0.624 | 0.014 | 0.524 | 0.201 | 0.577 | 0.612 | 0.791 | 0.608 | 0.840 |
| JSD | 0.285 | | 0.574 | | -0.102 | | 0.054 | | 0.128 | | 0.688 | | 0.522 | |
| MAND | 0.429 | | 0.764 | | 0.620 | | **0.410** | | 0.391 | | **0.852** | | 0.900 | |
| MCD | 0.637 | 0.619 | 0.323 | 0.752 | 0.304 | 0.847 | -0.037 | 0.567 | 0.382 | 0.636 | 0.645 | 0.816 | 0.721 | 0.845 |
| MMD | 0.033 | | 0.507 | | -0.068 | | 0.061 | | 0.168 | | 0.658 | | 0.504 | |
| PNORM | -0.446 | | 0.180 | | 0.071 | | -0.034 | | 0.174 | | 0.116 | | 0.321 | |
| TCP | 0.568 | 0.558 | 0.234 | 0.538 | 0.145 | 0.554 | 0.289 | 0.518 | 0.203 | 0.526 | 0.154 | 0.553 | 0.242 | 0.608 |

Table 5: **Evaluation of the methods for AllTextDistilBERT**, computed on target data for 5 random seeds, and different shift types and intensities. Error rate estimation is assessed with the Spearman's rank correlation ($\rho$). Error detection is evaluated with AUROC (auc) and is only applicable to EC and 9 baselines. For a given dataset and task, the best result is in **bold** (higher is better). The variability in results is displayed in appendix.

| Method | airbnb | | cloth | | kick | | petfinder | | salary | | wine10 | | wine100 | |
| --- | --- | --- | --- | --- | --- | --- | --- | --- | --- | --- | --- | --- | --- | --- |
| | $\rho$ | auc | $\rho$ | auc | $\rho$ | auc | $\rho$ | auc | $\rho$ | auc | $\rho$ | auc | $\rho$ | auc |
| AC | **0.711** | 0.634 | 0.336 | 0.760 | 0.548 | 0.869 | 0.135 | 0.570 | 0.352 | 0.654 | 0.621 | 0.845 | 0.765 | 0.851 |
| ACSC | 0.663 | 0.635 | 0.477 | **0.764** | 0.466 | 0.869 | 0.242 | 0.570 | 0.359 | **0.655** | 0.648 | **0.847** | 0.846 | **0.854** |
| ATC | 0.558 | | 0.546 | | 0.641 | | 0.481 | | 0.329 | | 0.807 | | 0.895 | |
| CP | 0.269 | 0.612 | 0.556 | 0.718 | 0.667 | 0.561 | 0.429 | 0.565 | 0.385 | 0.624 | 0.740 | 0.758 | 0.735 | 0.839 |
| DC | 0.069 | 0.502 | 0.368 | 0.537 | -0.264 | 0.475 | 0.184 | 0.504 | 0.230 | 0.501 | 0.539 | 0.621 | 0.396 | 0.578 |
| DENS | 0.523 | 0.605 | 0.538 | 0.756 | 0.658 | **0.879** | 0.159 | 0.544 | **0.463** | 0.648 | 0.660 | 0.814 | 0.782 | 0.820 |
| DNN | -0.604 | 0.543 | 0.192 | 0.625 | -0.046 | 0.563 | 0.033 | 0.524 | -0.014 | 0.543 | 0.494 | 0.786 | 0.624 | 0.709 |
| DOC | 0.626 | | 0.682 | | 0.710 | | **0.536** | | 0.362 | | 0.816 | | 0.914 | |
| EC (ours) | 0.676 | **0.646** | **0.751** | 0.753 | 0.675 | 0.873 | 0.434 | **0.582** | 0.195 | 0.614 | 0.808 | 0.844 | **0.922** | 0.842 |
| ENRG | 0.325 | 0.585 | 0.345 | 0.688 | 0.263 | 0.740 | 0.323 | 0.535 | 0.174 | 0.570 | 0.617 | 0.814 | 0.702 | 0.836 |
| JSD | 0.172 | | 0.409 | | -0.401 | | 0.273 | | 0.146 | | 0.634 | | 0.432 | |
| MAND | 0.615 | | 0.679 | | **0.714** | | 0.460 | | 0.362 | | **0.819** | | 0.915 | |
| MCD | 0.708 | 0.626 | 0.415 | 0.758 | 0.451 | 0.868 | 0.185 | 0.569 | 0.408 | 0.647 | 0.643 | 0.838 | 0.746 | 0.841 |
| MMD | 0.085 | | 0.554 | | -0.476 | | 0.316 | | 0.292 | | 0.621 | | 0.430 | |
| PNORM | -0.399 | | 0.093 | | 0.311 | | 0.047 | | -0.064 | | 0.087 | | 0.308 | |
| TCP | 0.260 | 0.526 | 0.134 | 0.539 | -0.141 | 0.624 | 0.033 | 0.503 | 0.152 | 0.525 | -0.033 | 0.533 | 0.098 | 0.560 |

# H    VARIABILITY IN RESULTS

The variability in results for LateFuseBERT architecture is presented in Table 6.

Table 6: **Variability in the results for LateFuseBERT**: Standard deviation of $\rho$ and auc results, computed based on 30 bootstraps with fraction 70% from raw table results (i.e. across seeds, shift types and intensities).

| Method | airbnb | | cloth | | kick | | petfinder | | salary | | wine10 | | wine100 | |
| --- | --- | --- | --- | --- | --- | --- | --- | --- | --- | --- | --- | --- | --- | --- |
| | $\rho$ | auc | $\rho$ | auc | $\rho$ | auc | $\rho$ | auc | $\rho$ | auc | $\rho$ | auc | $\rho$ | auc |
| AC | 0.079 | 0.001 | 0.056 | 0.001 | 0.055 | 0.001 | 0.061 | 0.002 | 0.093 | 0.002 | 0.033 | 0.001 | 0.039 | 0.001 |
| ACSC | 0.083 | 0.001 | 0.065 | 0.001 | 0.056 | 0.001 | 0.058 | 0.002 | 0.097 | 0.002 | 0.026 | 0.001 | 0.080 | 0.001 |
| ATC | 0.094 | | 0.075 | | 0.068 | | 0.074 | | 0.091 | | 0.022 | | 0.017 | |
| CP | 0.081 | 0.001 | 0.075 | 0.001 | 0.057 | 0.001 | 0.059 | 0.002 | 0.093 | 0.001 | 0.036 | 0.001 | 0.032 | 0.001 |
| DC | 0.082 | 0.002 | 0.068 | 0.002 | 0.077 | 0.002 | 0.053 | 0.001 | 0.077 | 0.001 | 0.056 | 0.002 | 0.068 | 0.002 |
| DENS | 0.070 | 0.001 | 0.057 | 0.001 | 0.077 | 0.001 | 0.065 | 0.001 | 0.074 | 0.002 | 0.030 | 0.001 | 0.025 | 0.001 |
| DNN | 0.058 | 0.001 | 0.067 | 0.002 | 0.096 | 0.002 | 0.075 | 0.002 | 0.076 | 0.001 | 0.076 | 0.001 | 0.060 | 0.001 |
| DOC | 0.084 | | 0.046 | | 0.062 | | 0.070 | | 0.073 | | 0.017 | | 0.019 | |
| EC (ours) | 0.067 | 0.002 | 0.025 | 0.001 | 0.045 | 0.001 | 0.052 | 0.001 | 0.071 | 0.001 | 0.017 | 0.001 | 0.018 | 0.001 |
| ENRG | 0.079 | 0.001 | 0.070 | 0.001 | 0.081 | 0.002 | 0.074 | 0.001 | 0.077 | 0.002 | 0.051 | 0.002 | 0.045 | 0.001 |
| JSD | 0.080 | | 0.077 | | 0.066 | | 0.068 | | 0.080 | | 0.039 | | 0.067 | |
| MAND | 0.082 | | 0.044 | | 0.061 | | 0.071 | | 0.070 | | 0.017 | | 0.018 | |
| MCD | 0.077 | 0.001 | 0.056 | 0.001 | 0.055 | 0.001 | 0.060 | 0.002 | 0.083 | 0.002 | 0.032 | 0.001 | 0.039 | 0.001 |
| MMD | 0.080 | | 0.064 | | 0.073 | | 0.054 | | 0.082 | | 0.051 | | 0.068 | |
| PNORM | 0.084 | | 0.070 | | 0.067 | | 0.077 | | 0.075 | | 0.091 | | 0.091 | |
| TCP | 0.082 | 0.001 | 0.061 | 0.002 | 0.059 | 0.002 | 0.066 | 0.002 | 0.086 | 0.001 | 0.085 | 0.002 | 0.079 | 0.001 |

Table 7: **Variability in the results for AllTextBERT**: Standard deviation of $\rho$ and auc results, computed based on 30 bootstraps with fraction 70% from raw table results (i.e. across seeds, shift types and intensities).

| Method | airbnb | | cloth | | kick | | petfinder | | salary | | wine10 | | wine100 | |
|---|---|---|---|---|---|---|---|---|---|---|---|---|---|---|
| | $\rho$ | auc | $\rho$ | auc | $\rho$ | auc | $\rho$ | auc | $\rho$ | auc | $\rho$ | auc | $\rho$ | auc |
| AC | 0.080 | 0.002 | 0.083 | 0.001 | 0.068 | 0.001 | 0.062 | 0.001 | 0.068 | 0.001 | 0.042 | 0.001 | 0.032 | 0.001 |
| ACSC | 0.093 | 0.002 | 0.081 | 0.001 | 0.068 | 0.001 | 0.062 | 0.001 | 0.056 | 0.001 | 0.075 | 0.001 | 0.069 | 0.001 |
| ATC | 0.108 | | 0.065 | | 0.040 | | 0.068 | | 0.076 | | 0.029 | | 0.019 | |
| CP | 0.066 | 0.002 | 0.062 | 0.001 | 0.076 | 0.002 | 0.055 | 0.001 | 0.086 | 0.001 | 0.030 | 0.002 | 0.046 | 0.001 |
| DC | 0.093 | 0.002 | 0.057 | 0.002 | 0.084 | 0.002 | 0.052 | 0.002 | 0.089 | 0.002 | 0.063 | 0.001 | 0.065 | 0.001 |
| DENS | 0.084 | 0.002 | 0.046 | 0.001 | 0.080 | 0.001 | 0.070 | 0.001 | 0.073 | 0.001 | 0.033 | 0.001 | 0.034 | 0.001 |
| DNN | 0.080 | 0.001 | 0.076 | 0.001 | 0.070 | 0.002 | 0.081 | 0.001 | 0.056 | 0.002 | 0.054 | 0.001 | 0.058 | 0.002 |
| DOC | 0.104 | | 0.047 | | 0.044 | | 0.062 | | 0.073 | | 0.022 | | 0.020 | |
| EC (ours) | 0.094 | 0.002 | 0.022 | 0.001 | 0.041 | 0.001 | 0.055 | 0.002 | 0.082 | 0.002 | 0.025 | 0.001 | 0.021 | 0.001 |
| ENRG | 0.073 | 0.002 | 0.065 | 0.001 | 0.087 | 0.001 | 0.058 | 0.001 | 0.069 | 0.001 | 0.058 | 0.001 | 0.032 | 0.001 |
| JSD | 0.084 | | 0.049 | | 0.085 | | 0.056 | | 0.087 | | 0.055 | | 0.068 | |
| MAND | 0.104 | | 0.048 | | 0.044 | | 0.067 | | 0.073 | | 0.022 | | 0.020 | |
| MCD | 0.085 | 0.002 | 0.080 | 0.001 | 0.070 | 0.001 | 0.063 | 0.001 | 0.065 | 0.001 | 0.048 | 0.001 | 0.030 | 0.001 |
| MMD | 0.094 | | 0.055 | | 0.092 | | 0.047 | | 0.081 | | 0.058 | | 0.070 | |
| PNORM | 0.082 | | 0.068 | | 0.058 | | 0.078 | | 0.068 | | 0.079 | | 0.088 | |
| TCP | 0.096 | 0.002 | 0.066 | 0.002 | 0.091 | 0.002 | 0.082 | 0.001 | 0.085 | 0.002 | 0.079 | 0.002 | 0.075 | 0.001 |

The variability in results for AllTextBERT architecture is presented in Table 7.

Table 8: **Variability in the results for LateFuseDistilBERT**: Standard deviation of $\rho$ and auc results, computed based on 30 bootstraps with fraction 70% from raw table results (i.e. across seeds, shift types and intensities).

| Method | airbnb | | cloth | | kick | | petfinder | | salary | | wine10 | | wine100 | |
|---|---|---|---|---|---|---|---|---|---|---|---|---|---|---|
| | $\rho$ | auc | $\rho$ | auc | $\rho$ | auc | $\rho$ | auc | $\rho$ | auc | $\rho$ | auc | $\rho$ | auc |
| AC | 0.078 | 0.002 | 0.066 | 0.001 | 0.086 | 0.001 | 0.074 | 0.002 | 0.089 | 0.002 | 0.046 | 0.001 | 0.042 | 0.001 |
| ACSC | 0.078 | 0.002 | 0.072 | 0.001 | 0.077 | 0.001 | 0.075 | 0.002 | 0.082 | 0.002 | 0.044 | 0.001 | 0.034 | 0.001 |
| ATC | 0.089 | | 0.050 | | 0.069 | | 0.074 | | 0.092 | | 0.025 | | 0.021 | |
| CP | 0.080 | 0.001 | 0.030 | 0.001 | 0.050 | 0.002 | 0.074 | 0.002 | 0.086 | 0.001 | 0.029 | 0.002 | 0.035 | 0.001 |
| DC | 0.076 | 0.001 | 0.055 | 0.001 | 0.073 | 0.002 | 0.070 | 0.001 | 0.090 | 0.001 | 0.052 | 0.002 | 0.061 | 0.002 |
| DENS | 0.066 | 0.001 | 0.040 | 0.001 | 0.083 | 0.001 | 0.070 | 0.001 | 0.076 | 0.001 | 0.030 | 0.001 | 0.033 | 0.001 |
| DNN | 0.085 | 0.002 | 0.087 | 0.002 | 0.076 | 0.002 | 0.082 | 0.002 | 0.071 | 0.002 | 0.067 | 0.001 | 0.067 | 0.001 |
| DOC | 0.075 | | 0.036 | | 0.072 | | 0.072 | | 0.081 | | 0.018 | | 0.016 | |
| EC (ours) | 0.073 | 0.001 | 0.030 | 0.001 | 0.056 | 0.001 | 0.074 | 0.002 | 0.082 | 0.002 | 0.019 | 0.001 | 0.014 | 0.001 |
| ENRG | 0.079 | 0.002 | 0.075 | 0.001 | 0.086 | 0.001 | 0.094 | 0.002 | 0.076 | 0.001 | 0.048 | 0.001 | 0.067 | 0.001 |
| JSD | 0.083 | | 0.057 | | 0.063 | | 0.081 | | 0.096 | | 0.047 | | 0.078 | |
| MAND | 0.084 | | 0.035 | | 0.072 | | 0.073 | | 0.082 | | 0.018 | | 0.016 | |
| MCD | 0.078 | 0.002 | 0.065 | 0.001 | 0.083 | 0.001 | 0.071 | 0.002 | 0.082 | 0.002 | 0.044 | 0.001 | 0.044 | 0.001 |
| MMD | 0.074 | | 0.054 | | 0.071 | | 0.071 | | 0.091 | | 0.048 | | 0.067 | |
| PNORM | 0.100 | | 0.068 | | 0.092 | | 0.079 | | 0.082 | | 0.082 | | 0.073 | |
| TCP | 0.083 | 0.001 | 0.079 | 0.001 | 0.094 | 0.002 | 0.059 | 0.002 | 0.078 | 0.001 | 0.081 | 0.001 | 0.072 | 0.002 |

The variability in results for LateFuseDistilBERT architecture is presented in Table 8.

Table 9: **Variability in the results for AllTextDistilBERT**: Standard deviation of $\rho$ and auc results, computed based on 30 bootstraps with fraction 70% from raw table results (i.e. across seeds, shift types and intensities).

| Method | airbnb | | cloth | | kick | | petfinder | | salary | | wine10 | | wine100 | |
|---|---|---|---|---|---|---|---|---|---|---|---|---|---|---|
| | $\rho$ | auc | $\rho$ | auc | $\rho$ | auc | $\rho$ | auc | $\rho$ | auc | $\rho$ | auc | $\rho$ | auc |
| AC | 0.042 | 0.001 | 0.092 | 0.001 | 0.072 | 0.001 | 0.088 | 0.002 | 0.064 | 0.001 | 0.068 | 0.001 | 0.036 | 0.001 |
| ACSC | 0.044 | 0.001 | 0.073 | 0.001 | 0.071 | 0.001 | 0.073 | 0.002 | 0.064 | 0.001 | 0.065 | 0.001 | 0.021 | 0.001 |
| ATC | 0.066 | | 0.051 | | 0.060 | | 0.068 | | 0.081 | | 0.036 | | 0.018 | |
| CP | 0.074 | 0.002 | 0.049 | 0.001 | 0.043 | 0.001 | 0.057 | 0.001 | 0.076 | 0.001 | 0.033 | 0.001 | 0.054 | 0.001 |
| DC | 0.101 | 0.001 | 0.060 | 0.001 | 0.076 | 0.002 | 0.048 | 0.001 | 0.088 | 0.001 | 0.054 | 0.002 | 0.065 | 0.002 |
| DENS | 0.066 | 0.002 | 0.063 | 0.001 | 0.056 | 0.001 | 0.082 | 0.002 | 0.082 | 0.001 | 0.044 | 0.001 | 0.032 | 0.001 |
| DNN | 0.047 | 0.002 | 0.075 | 0.002 | 0.072 | 0.003 | 0.073 | 0.002 | 0.076 | 0.001 | 0.053 | 0.002 | 0.057 | 0.002 |
| DOC | 0.052 | | 0.045 | | 0.055 | | 0.065 | | 0.072 | | 0.034 | | 0.015 | |
| EC (ours) | 0.055 | 0.001 | 0.030 | 0.001 | 0.056 | 0.001 | 0.082 | 0.001 | 0.074 | 0.001 | 0.028 | 0.001 | 0.015 | 0.001 |
| ENRG | 0.093 | 0.002 | 0.073 | 0.001 | 0.075 | 0.001 | 0.068 | 0.002 | 0.064 | 0.002 | 0.058 | 0.001 | 0.050 | 0.001 |
| JSD | 0.094 | | 0.066 | | 0.069 | | 0.053 | | 0.095 | | 0.049 | | 0.070 | |
| MAND | 0.051 | | 0.045 | | 0.055 | | 0.064 | | 0.074 | | 0.034 | | 0.015 | |
| MCD | 0.050 | 0.002 | 0.078 | 0.001 | 0.077 | 0.001 | 0.083 | 0.002 | 0.066 | 0.001 | 0.066 | 0.001 | 0.039 | 0.001 |
| MMD | 0.100 | | 0.056 | | 0.077 | | 0.050 | | 0.078 | | 0.053 | | 0.068 | |
| PNORM | 0.086 | | 0.081 | | 0.087 | | 0.095 | | 0.072 | | 0.079 | | 0.060 | |
| TCP | 0.086 | 0.002 | 0.079 | 0.001 | 0.090 | 0.002 | 0.062 | 0.002 | 0.071 | 0.001 | 0.065 | 0.002 | 0.088 | 0.001 |

The variability in results for AllTextDistilBERT architecture is presented in Table 9.

# I    ABLATION STUDIES

We also perform ablation studies to compare the results of EC with (1) Ablation 1: an Error Classifier using only the [CLS] tokens' output embeddings as features, or (2) Ablation 2: EC using only the classifier's output $\hat{\pi}(\mathbf{x})$ as features. The results for LateFuseBERT, AllTextBERT, LateFuseDistilBERT, and AllTextDistilBERT architectures are presented in Tables 10, 11, 12, and 13, respectively. The results show that Ablation 2 performs best for unsupervised performance estimation (first rank in 17 use cases out of 28), whereas EC performs best for error detection (first rank in 14 use cases out of 28). Further, EC seems to be more stable overall, while Ablation 2's performance is significantly lower for some of the use cases: for example, on cloth for $\rho$ (LateFuseBERT and AllTextDistilBERT), or on kick for auc (AllTextBERT).

Table 10: **Results of the ablation study for LateFuseBERT**.

| Method | airbnb | | cloth | | kick | | petfinder | | salary | | wine10 | | wine100 | |
|---|---|---|---|---|---|---|---|---|---|---|---|---|---|---|
| | $\rho$ | auc | $\rho$ | auc | $\rho$ | auc | $\rho$ | auc | $\rho$ | auc | $\rho$ | auc | $\rho$ | auc |
| Ablation 1 | 0.150 | 0.625 | **0.821** | **0.757** | **0.850** | **0.885** | **0.555** | **0.611** | 0.465 | 0.621 | 0.890 | 0.833 | 0.902 | 0.834 |
| Ablation 2 | **0.282** | **0.641** | 0.699 | 0.720 | 0.753 | 0.766 | 0.531 | 0.552 | **0.490** | 0.593 | **0.915** | **0.844** | **0.933** | **0.858** |
| EC | 0.203 | 0.632 | 0.804 | 0.755 | 0.835 | **0.885** | 0.523 | 0.608 | 0.450 | **0.622** | 0.904 | 0.843 | 0.927 | 0.857 |

Table 11: **Results of the ablation study for AllTextBERT**.

| Method | airbnb | | cloth | | kick | | petfinder | | salary | | wine10 | | wine100 | |
|---|---|---|---|---|---|---|---|---|---|---|---|---|---|---|
| | $\rho$ | auc | $\rho$ | auc | $\rho$ | auc | $\rho$ | auc | $\rho$ | auc | $\rho$ | auc | $\rho$ | auc |
| Ablation 1 | 0.423 | **0.633** | **0.794** | 0.754 | 0.755 | **0.877** | 0.313 | 0.567 | 0.237 | **0.617** | 0.893 | 0.840 | 0.831 | 0.818 |
| Ablation 2 | 0.365 | 0.610 | 0.747 | 0.716 | 0.664 | 0.782 | **0.374** | 0.538 | **0.273** | 0.582 | **0.912** | 0.844 | **0.914** | 0.846 |
| EC | **0.425** | **0.633** | 0.789 | **0.756** | **0.759** | **0.877** | 0.360 | **0.571** | 0.236 | 0.615 | 0.903 | **0.849** | 0.907 | **0.847** |

Table 12: **Results of the ablation study for LateFuseDistilBERT**.

| Method | airbnb | | cloth | | kick | | petfinder | | salary | | wine10 | | wine100 | |
|--------|--------|--------|--------|--------|--------|--------|--------|--------|--------|--------|--------|--------|--------|--------|
| | $\rho$ | auc | $\rho$ | auc | $\rho$ | auc | $\rho$ | auc | $\rho$ | auc | $\rho$ | auc | $\rho$ | auc |
| Ablation 1 | 0.432 | 0.626 | 0.796 | **0.763** | **0.740** | 0.878 | 0.379 | 0.594 | 0.338 | **0.619** | 0.773 | 0.819 | 0.830 | 0.821 |
| Ablation 2 | **0.566** | 0.633 | **0.810** | 0.724 | 0.688 | 0.760 | 0.344 | 0.534 | **0.503** | 0.603 | **0.851** | **0.837** | **0.924** | **0.850** |
| EC | 0.468 | **0.636** | 0.791 | 0.761 | 0.718 | **0.879** | **0.390** | **0.597** | 0.366 | 0.618 | 0.830 | 0.836 | 0.912 | 0.849 |

Table 13: **Results of the ablation study for AllTextDistilBERT**.

| Method | airbnb | | cloth | | kick | | petfinder | | salary | | wine10 | | wine100 | |
|--------|--------|--------|--------|--------|--------|--------|--------|--------|--------|--------|--------|--------|--------|--------|
| | $\rho$ | auc | $\rho$ | auc | $\rho$ | auc | $\rho$ | auc | $\rho$ | auc | $\rho$ | auc | $\rho$ | auc |
| Ablation 1 | 0.620 | 0.641 | 0.748 | **0.754** | 0.669 | **0.874** | 0.439 | 0.581 | 0.200 | **0.617** | 0.795 | 0.830 | 0.852 | 0.808 |
| Ablation 2 | 0.590 | 0.618 | 0.640 | 0.719 | 0.624 | 0.778 | **0.446** | 0.556 | **0.343** | 0.595 | **0.836** | 0.843 | **0.924** | **0.844** |
| EC | **0.676** | **0.646** | **0.751** | 0.753 | **0.675** | 0.873 | 0.434 | **0.582** | 0.195 | 0.614 | 0.808 | **0.844** | 0.922 | 0.842 |

## J  IMPLEMENTATION AND COMPUTATIONAL INFORMATION

**Hardware and computational cost.** We run the experiments with a Tesla T4 GPU. Table 14 summarizes the average computational cost for each method (EC and all the baselines). The methods that require performing several forward passes during inference (e.g. MCD), training one or several models (e.g. DENS, EC), or computing the source error rate (DOC) are less efficient than the other baselines.

Table 14: Average computation time (in seconds) computed for each method, over various model architectures, dataset seeds, shift types and intensities.

| Method | airbnb | cloth | kick | petfinder | salary | wine10 | wine100 |
|--------|--------|-------|------|-----------|--------|--------|---------|
| AC | 0.01 | 0.01 | 0.01 | 0.01 | 0.01 | 0.01 | 0.01 |
| ACSC | 0.01 | 0.01 | 0.01 | 0.01 | 0.01 | 0.01 | 0.01 |
| ATC | 0.01 | 0.03 | 0.03 | 0.02 | 0.03 | 0.03 | 0.04 |
| CP | 0.01 | 0.01 | 0.01 | 0.01 | 0.01 | 0.01 | 0.01 |
| DC | 1.98 | 9.20 | 4.34 | 4.29 | 5.49 | 5.03 | 4.69 |
| DENS | 3.59 | 11.03 | 15.87 | 7.39 | 8.67 | 15.62 | 15.72 |
| DNN | 0.12 | 0.31 | 0.42 | 0.22 | 0.24 | 0.45 | 0.40 |
| DOC | 18.99 | 17.39 | 12.45 | 18.90 | 5.73 | 16.15 | 15.87 |
| EC (ours) | 4.22 | 15.83 | 17.12 | 9.86 | 11.41 | 21.70 | 25.72 |
| ENRG | 0.00 | 0.00 | 0.00 | 0.00 | 0.00 | 0.00 | 0.00 |
| JSD | 0.01 | 0.01 | 0.01 | 0.01 | 0.01 | 0.01 | 0.01 |
| MAND | 0.01 | 0.01 | 0.01 | 0.01 | 0.01 | 0.01 | 0.01 |
| MCD | 88.37 | 24.92 | 12.66 | 40.53 | 10.71 | 16.05 | 15.95 |
| MMD | 0.10 | 0.46 | 0.81 | 0.26 | 0.32 | 0.82 | 0.83 |
| PNORM | 1.38 | 2.87 | 3.84 | 2.13 | 2.42 | 3.80 | 3.80 |
| TCP | 0.76 | 2.31 | 3.30 | 1.54 | 1.81 | 3.25 | 3.24 |

**Python libraries.** The implementation is based on Python 3.10 and the following packages: torch 2.4.0+cu121, transformers 4.42.4, scikit-learn 1.3.2, scipy 1.13.1, pandas 2.1.4, numpy 1.26.4, ATC (https://github.com/saurabhgarg1996/ATC_code), Mandoline (https://github.com/HazyResearch/mandoline), typo (https://github.com/ranvijaykumar/typo), matplotlib 3.7.1, and seaborn 0.13.1. These libraries are publicly available with "BSD", "MIT", or "Apache Software" licenses.

## K EXPLANATION ALGORITHM: A FEW EXAMPLES

### K.1 EXAMPLE 1: TYPOGRAPHICAL ERROR (CLOTH, ALLTEXTDISTILBERT)

The example is the same as the one presented in Figure 1 where LateFuseDistilBERT was the classification model. The only difference here is that AllTextDistilBERT is used as classification model. The multimodal input is described below. Unlike AllTextDistilBERT, the model still predicts the correct label after implementing the shift. The resulting top 10 contributions from Figure 5 (left) show the uncertainty introduced by the token "flat" (positive contribution to probability of failure). However, the positive contributions are less important than with LateFuseDistilBERT (Figure 1) and EC value remains lower too (45% vs 63% with LateFuseDistilBERT). There is no significant disagreement between the various explanation methods, as demonstrated in Figure 5 (right).

- Dataset: cloth.

- Classification model: AllTextDistilBERT.

- Shift type: typo.

- Shift intensity: 1 typo ('Unflattering': 'Un flattering').

- Categorical variables: Division Name ('General'), Department Name ('Dresses'), Class Name ('Dresses').

- Numerical variables: Age (55), Positive Feedback Count (0).

- Text field: "Un flattering I purchased the blue with white dots. the shape was awful, but looked like a sack - returned".

- Text field (after early fusion): "Division Name General Department Name Dresses Class Name Dresses Age 55 Positive Feedback Count 0 Un flattering I purchased the blue with white dots. the shape was awful, but looked like a sack - returned".

- True label (rating): 2.

- Predicted label: 2.

- EC value: 45% (31% without typo).

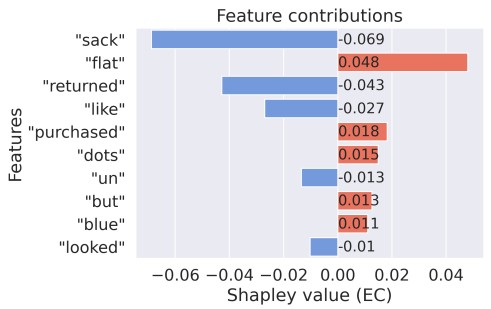
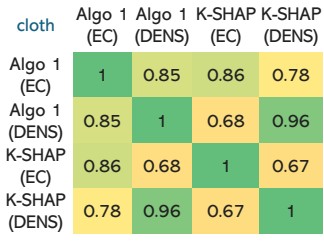

Figure 5: **Typographical error (cloth, AllTextDistilBERT).** *Left*: Top 10 feature contributions (Algorithm 1: EC as value function). *Right*: Pearson correlation matrix between the outputs of various explanation methods.

### K.2 EXAMPLE 2: *emptyCategory_typos* (WINE10, LATEFUSEBERT)

In this example of multimodal shift, EC predicts a quite high likelihood of error (59%). The explanation (Figure 6 (left)) clearly shows that empty categorical values (country and year) produce uncertainties. Further, as the typos affect keywords, this makes the classification task more difficult. Based on the correlation matrix (Figure 6 (right)), the Kernel SHAP algorithm based on EC values would be more trustworthy.

- Dataset: wine10.

- Classification model: LateFuseBERT.

- Shift type: *emptyCategory_typos*.

- Shift intensity: 25 typos, $50\%$ of rows affected by empty categorical values.

- Categorical variables: country (" "), year (" ").

- Numerical variables: points (89), price (35.0).

- Text field: "Made in adrie, leaner adn earthider styl e, w th herb a nd tobacco-nf used lbackberry, currant and chedryflavors.A touch of etylish oak adds a fisnenote. Not a blockuster, but balaend and elgeant. Rrink now.".

- True label: "Merlot".

- Predicted label: "Cabernet Sauvignon".

- EC value: $59\%$ ($19\%$ is the average on $\mathcal{S}_{val}$).

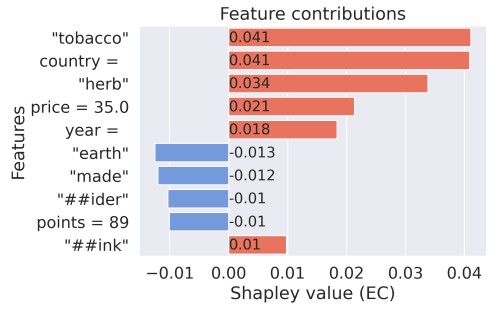 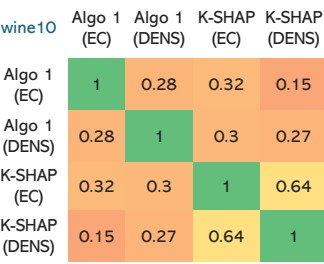

Figure 6: **emphemptyCategory_typos (wine10, LateFuseBERT).** *Left*: Top 10 feature contributions (Algorithm 1: EC as value function). *Right*: Pearson correlation matrix between the outputs of various explanation methods.

### K.3   EXAMPLE 3: *newClass* (PETFINDER, LATEFUSEBERT)

We describe an example of out-of-domain shift. EC predicts a high probability of error ($89\%$) and the top 10 contributions are all positive (Figure 7). The outputs from Algorithm 1 are in agreement (EC and DENS as value functions), but they disagree with the outcomes of the Kernel SHAP algorithm. Therefore, the explanations provided here might be less reliable.

- Dataset: petfinder.

- Classification model: LateFuseBERT.

- Shift type: *newClass*.

- Categorical variables: Type (1="Dog"), Breed1 (205="Shih Tzu"), Gender (2="Female"), Color1 (3="Golden"), MaturitySize (1="Small"), FurLength (2="Medium") (...).

- Numerical variables: Age (54 months), Quantity (1), Fee (350) (...).

- Text field: "She is very quiet and a very good watch dog she can get along we'll with other dogs . Friendly , easy going loves to watch tv send me SMS if u interested pls no calls thank you . Reason giving away too many dogs at home".

- True label: 0 (Lower is faster).

- Predicted label: 4.

- EC value: $89\%$ ($61\%$ is the average on $\mathcal{S}_{val}$).

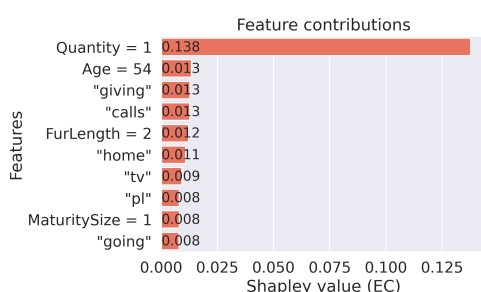
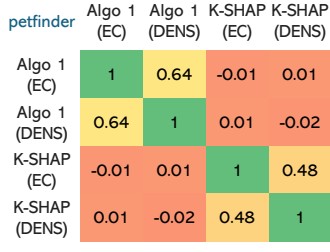

Figure 7: *newClass* **(petfinder, LateFuseBERT).** *Left*: Top 10 feature contributions (Algorithm 1: EC as value function). *Right*: Pearson correlation matrix between the outputs of various explanation methods.

## L    EC SCATTER PLOTS FOR LATEFUSEBERT

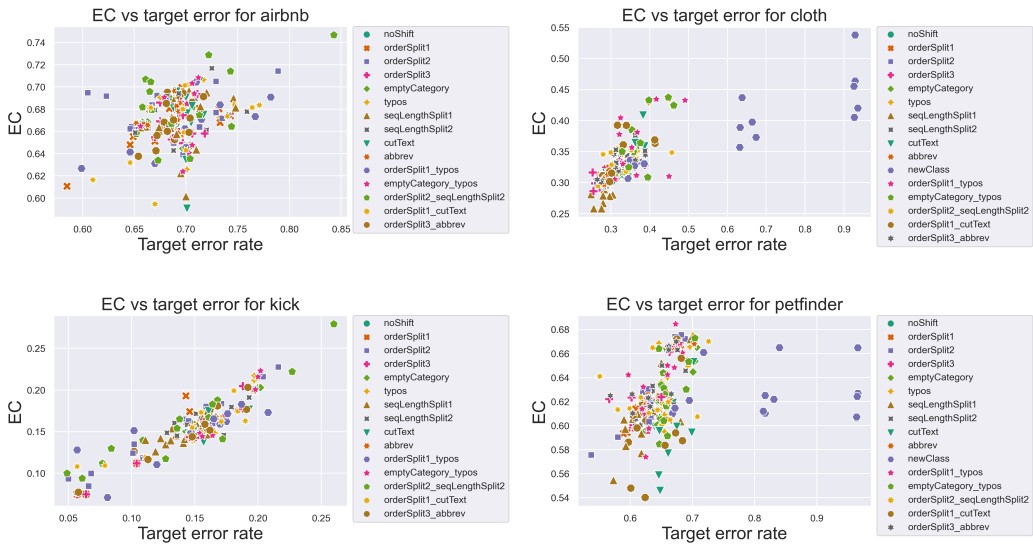

Figure 8: From top to bottom , left to right. **Score versus true error rate** for the task of unsupervised performance estimation, for EC, on airbnb/cloth/kick/petfinder target data (with LateFuseBERT), by shift type of various intensities and different seeds. *orderSplit1/2/3* correspond to 3 different tabular features affected by the shift, while *seqLengthSplit1/2* correspond to ascending/descending order, respectively.

For the task of unsupervised performance estimation with EC, we display the scatter plots (score versus true error rate) with LateFuseBERT as classification model on the following target datasets in Figure 8: airbnb, cloth, kick, petfinder. The performance depends on the use case and type of shifts. For some of the datasets, the data points related to *newClass* appear as outliers that are more difficult to assess.

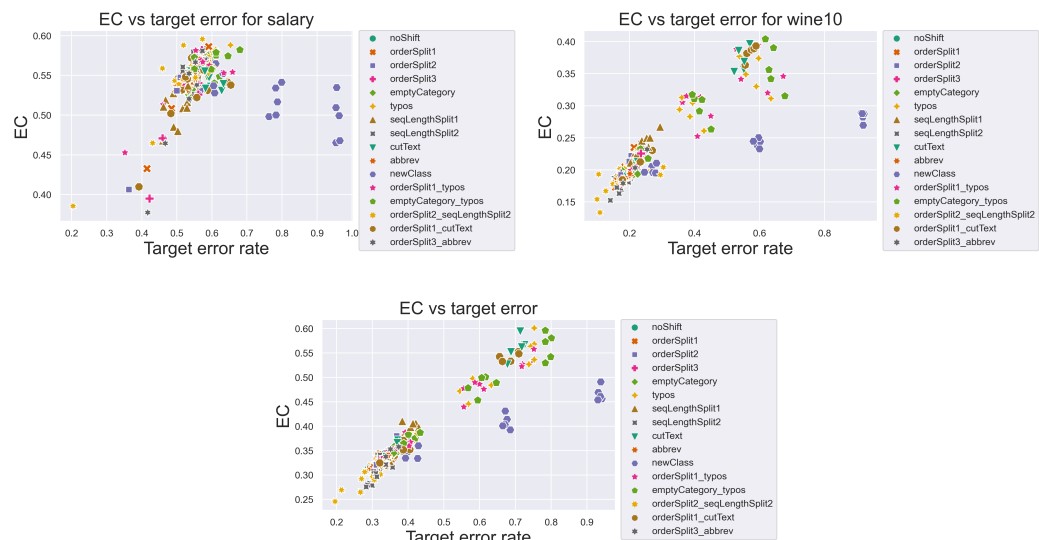

Figure 9: From top to bottom , left to right. **Score versus true error rate** for the task of unsupervised performance estimation, for EC, on salary/wine10/wine100 target data (with LateFuseBERT), by shift type of various intensities and different seeds. *orderSplit1/2/3* correspond to 3 different tabular features affected by the shift, while *seqLengthSplit1/2* correspond to ascending/descending order, respectively.

For the task of unsupervised performance estimation with EC, we display the scatter plots (score versus true error rate) with LateFuseBERT as classification model on the following target datasets in Figure 9: salary, wine10, wine100. The performance depends on the use case and type of shifts. For some of the datasets, the data points related to *newClass* appear as outliers that are more difficult to assess.

