# OpenReview forum: "Evaluating and Explaining the Severity of Distribution Shifts: Illustration with Tabular Text Classification"
_ICLR.cc/2025/Conference — ICLR 2025 Conference Withdrawn Submission_

### Official Review · Reviewer_v4zh · 2024-11-02

**Soundness:** 2
**Presentation:** 2
**Contribution:** 2
**Rating:** 3
**Confidence:** 3

**Summary:**

In this paper, the authors introduce the Error Classifier which outputs the probability that a given classification model will fail based on the detected error patterns. Then they employ a sampling-based approximation of Shapley values to explain why a shift is predicted as severe, in terms of feature values, that is, they focus on investigating what feature values contribute to the likelihood of failure assessed by Error Classifier. In general, this paper provides a new method to quantitively analyze what feature values contribute to the likelihood of failure assessed by Error Classifier.

**Strengths:**

The authors did extensive experiments to demonstrate the advantages of the proposed method.

**Weaknesses:**

The contribution of this paper is limited. There are not any theory to support the proposed method. The writing in methodology part especially in Section 3.3 is unclear.

**Questions:**

1, The proposed error classifier may not be robust because we need to make sure both $\hat{\pi}$ and $\hat{f}$ are correctly specified. If we specify a wrong model in estimation process of $\hat{\pi}$ or $\hat{f}$, what will happen? In practice, it may be very hard to correctly specify the model, so the authors need to demonstrate the robustness of the proposed method to the model misspecification.
2, How to learn feature encoder $\varphi$? I do not understand why the authors do not consider the feature encoder when training $\hat{\pi}$ but consider the feature encoder when training $\hat{f}$.
3, In line 7 of Algorithm 1, the readers may want to know what [MASK] is?
4, I do not understand why the implementation in Algorithm 1 can reflect the contribution of a tabular feature with index j or a text feature (i.e. token) with index (i.e. position) j. This is not intuitively obvious. If the authors can give more explanation, it will make this paper more clear.

---

### Official Review · Reviewer_V4Cj · 2024-11-04

**Soundness:** 3
**Presentation:** 2
**Contribution:** 2
**Rating:** 3
**Confidence:** 5

**Summary:**

The paper addresses the task of evaluating and explaining model performance under distribution shifts. To this end, the authors introduce EC, a method for unsupervised performance estimation and error detection. They use Shapley values to explain why shifts are predicted as severe and apply their method on tabular data.

**Strengths:**

The authors test 2 multimodal fusion strategies for classification models and propose ways to evaluate explanation consistency across different methods. They evaluate their methods on 7 different datasets.

**Weaknesses:**

The scope of the proposed method is very limited with a focus on tabular-text data and the approach should also be evaluated on other modalities (image especially).

In terms of the empirical evaluation, a key weakness is the evaluation on synthetic shifts only - it is unclear whether the proposed results would translate to real world settings.
In terms of baselines, the authors focus solely on unsupervised performance estimation. They have missed a recently proposed method for explanatory model performance estimation that is tailored to the same task [1]. In that work, the authors apply their method on different data modalities (see comment above) and also introduce metrics to quantify the quality of the explanations. A thorough comparison to this work is crucial.
The proposed method is also purely empirical and lacks theoretical guarantees for performance bounds (in contrasted eg to the missed baseline).


[1] Decker, Thomas, et al. "Explanatory Model Monitoring to Understand the Effects of Feature Shifts on Performance." Proceedings of the 30th ACM SIGKDD Conference on Knowledge Discovery and Data Mining. 2024.

**Questions:**

See above

---

### Official Review · Reviewer_s1Er · 2024-11-05

**Soundness:** 2
**Presentation:** 2
**Contribution:** 2
**Rating:** 3
**Confidence:** 3

**Summary:**

The authors tackle the problem of error prediction and explanation under distribution shift. They propose training an error classifier on a heldout dataset, which directly predicts whether the original network will make an error, and which can be used to infer whether the model will make an error on the test dataset. Running SHAP on this error classifier also allows the user to get an explanation of why the original model made a mistake. The authors evaluate their method on multimodal datasets consisting of a tabular + text modality, finding that they outperform the baselines on error detection.

**Strengths:**

- The paper tackles an important real-world problem.
- The authors compare against a large and comprehensive grid of baselines.

**Weaknesses:**

1. The significance of the paper is greatly diminished by the fact that the authors only study multimodal datasets consisting of tabular + text modalities. It seems to me that the method can easily be applied to many other settings (e.g. unimodal, images). Why do the authors restrict themselves to this setting? The authors should consider evaluating their method on additional distribution shift setups, e.g. in DomainBed or WILDS [1-2].

2. The idea of an error classifier (i.e. training a model to predict the loss of another model) is not new, and has been well-studied in prior work [3-4].

3. Currently, the evaluation of explanations (e.g. Figure 3) is quite ad-hoc and unsystematic. The authors should instead introduce targeted shifts on important features, which they can then use as the gold standard to see the ranking in their explanations.

4. One major issue in the error classifier is that it also experiences a distribution shift between the dataset where it is trained and where it is evaluated. As such, there are no guarantees that the error classifier will be accurate for all possible shifts, or that the explanations will be reasonable. At a minimum, there is some assumption of overlapping support required.

5. The utility of Algorithm 1 is unclear to me. How is it different from running Kernel SHAP on the error classifier? Does Algorithm 1 output Shapley values that satisfy the typical properties (e.g. efficiency, symmetry, etc)?

[1] In Search of Lost Domain Generalization. ICLR 2021.

[2] WILDS: A Benchmark of in-the-Wild Distribution Shifts. ICML 2021.

[3] Learning Loss for Active Learning. CVPR 2019.

[4] Learning Loss for Test-Time Augmentation. NeurIPS 2020.

**Questions:**

1. The authors should comment on the connections between the error classifier, and epistemic and aleatoric uncertainty. Would the EC only be able to detect errors due to epistemic uncertainty, or both types?

---

### Note · Authors · 2024-11-12

I have read and agree with the venue's withdrawal policy on behalf of myself and my co-authors.